# Variational Monte Carlo on a Budget –
# Fine-tuning pre-trained Neural Wavefunctions

**Michael Scherbela**[*]
University of Vienna
michael.scherbela@univie.ac.at

**Leon Gerard**[*]
University of Vienna
leon.gerard@univie.ac.at

**Philipp Grohs**
University of Vienna
philipp.grohs@univie.ac.at

## Abstract

Obtaining accurate solutions to the Schrödinger equation is the key challenge in computational quantum chemistry. Deep-learning-based Variational Monte Carlo (DL-VMC) has recently outperformed conventional approaches in terms of accuracy, but only at large computational cost. Whereas in many domains models are trained once and subsequently applied for inference, accurate DL-VMC so far requires a full optimization for every new problem instance, consuming thousands of GPUhs even for small molecules. We instead propose a DL-VMC model which has been pre-trained using self-supervised wavefunction optimization on a large and chemically diverse set of molecules. Applying this model to new molecules without any optimization, yields wavefunctions and absolute energies that outperform established methods such as CCSD(T)-2Z. To obtain accurate relative energies, only few fine-tuning steps of this base model are required. We accomplish this with a fully end-to-end machine-learned model, consisting of an improved geometry embedding architecture and an existing SE(3)-equivariant model to represent molecular orbitals. Combining this architecture with continuous sampling of geometries, we improve zero-shot accuracy by two orders of magnitude compared to the state of the art. We extensively evaluate the accuracy, scalability and limitations of our base model on a wide variety of test systems.

## 1 Introduction

Solving the Schrödinger equation is of utmost importance for the prediction of quantum chemical properties in chemistry. The time-independent Schrödinger equation in the Born-Oppenheimer approximation [1] for a molecule with $N_{\text{nuc}}$ nuclei and $n_{\text{el}}$ electrons is an eigenvalue problem with Hamiltonian $H$:

$$H\psi = E\psi, \qquad H = -\frac{1}{2}\sum_i \nabla^2_{\boldsymbol{r}_i} + \sum_{i>j}\frac{1}{r_{ij}} + \sum_{I>J}\frac{Z_I Z_J}{R_{IJ}} - \sum_{i,I}\frac{Z_I}{r_{iI}}. \tag{1}$$

By $\boldsymbol{R} = (\boldsymbol{R}_1, \ldots, \boldsymbol{R}_{N_{\text{nuc}}}) \in \mathbb{R}^{N_{\text{nuc}} \times 3}$ and $\boldsymbol{Z} = (Z_1, \ldots, Z_{N_{\text{nuc}}}) \in \mathbb{N}^{N_{\text{nuc}}}$ we denote the nuclear positions and charges. The electron positions are denoted by $\boldsymbol{r} = (\boldsymbol{r_1}, \ldots, \boldsymbol{r_{n_\uparrow}}, \ldots, \boldsymbol{r_{n_{\text{el}}}}) \in \mathbb{R}^{n_{\text{el}} \times 3}$ with $n_\uparrow$ spin-up electrons and $n_\downarrow$ spin-down electrons. The inter-particle difference and distance vectors are written as $\boldsymbol{R}_{IJ} = \boldsymbol{R}_I - \boldsymbol{R}_J$, $R_{IJ} = |\boldsymbol{R}_{IJ}|$, $\boldsymbol{r}_{iI} = \boldsymbol{r}_i - \boldsymbol{R}_I$, $r_{iI} = |\boldsymbol{r}_{iI}|$, $\boldsymbol{r}_{ij} = \boldsymbol{r}_i - \boldsymbol{r}_j$ and $r_{ij} = |\boldsymbol{r}_{ij}|$ with $I, J = 1, \ldots, N_{\text{nuc}}$ and $i, j = 1, \ldots, n_{\text{el}}$. The eigenvalues $E$ of Eq. 1 represent

---

[*]Equal contribution, author order random

the energy states of a molecule, whereas a special interest lies in finding the smallest eigenvalue $E_0$, called the ground-state energy. The corresponding high-dimensional wavefunction $\psi : \mathbb{R}^{n_{el} \times 3} \to \mathbb{R}$ can be found via the Rayleigh-Ritz principle, by minimizing

$$\mathcal{L}(\psi_\theta) = \mathbb{E}_{\boldsymbol{r} \sim \psi_\theta^2(\boldsymbol{r})} \left[ \frac{H\psi_\theta(\boldsymbol{r})}{\psi_\theta(\boldsymbol{r})} \right] \geq E_0. \tag{2}$$

Due to electrons being fermions, the solution must fulfill the anti-symmetry property, stating that the sign of the wavefunction must change for any permutation $\mathcal{P}$ of two electrons of the same spin: $\psi(\boldsymbol{r}) = -\psi(\mathcal{P}\boldsymbol{r})$. Having access to the solution $\psi$, allows in principle a complete description of the considered molecule. Unfortunately, only for one electron systems there exists an analytical solution and due to the curse of dimensionality, with increasing number of particles, obtaining an accurate approximation of the wavefunction becomes intractable already for medium-sized molecules. This is because many high-accuracy approximation methods scale poorly with $n_{el}$. For example CCSD(T) – coupled cluster with its single-, double-, and perturbative triple-excitations variant – is considered the gold-standard reference in computational chemistry, but its computational cost scales as $\mathcal{O}(n_{el}^7)$ [2]. Deep-learning-based Variational Monte Carlo (DL-VMC) has emerged as a promising alternative solution. A single step scales as $\mathcal{O}(n_{el}^4)$ and it has surpassed the accuracy of many conventional methods such as CCSD(T), when applied to small molecules [3]. In DL-VMC, the wavefunction is represented by a neural network with trainable parameters $\theta$ and optimized via Eq. 2 using gradient based optimization. Since the expectation value of Eq. 2 cannot be computed analytically, it is approximated by sampling the electron positions during optimization and evaluation with Monte Carlo methods like Metropolis-Hastings [4].

**Related work**   FermiNet by Pfau et al. [3] and its variants have emerged as the leading architecture for DL-VMC in first quantization. It can reach highly accurate energies, but typically requires tens of thousands of optimization steps for convergence. Many improvements have been proposed to further increase accuracy [5–7] and accelerate convergence [8]. Furthermore, DL-VMC has been extended to properties beyond energies [9–11] and systems beyond molecules [12–16]. Despite the favorable scaling of DL-VMC, computational cost is still high, even for small molecules, often requiring thousands of GPUhs to find $\psi$ for a single small molecule [17]. This is because unlike typical machine learning applications – which train an expensive model once, and subsequently achieve cheap inference – in DL-VMC the minimization of Eq. 2 is typically done from scratch for every new system.

A promising line of research to scale-up expensive ab-initio solvers such as DL-VMC or CCSD(T) has been to develop proxy methods, which can be trained on outputs of ab-initio methods to directly predict molecular properties [18, 19] or wavefunctions [20, 21] from the molecular geometry. While these proxy methods can often reproduce the underlying ab-initio method with high fidelity and scale to millions of atoms [22], they are fundamentally limited by the accuracy of their reference method and need many high-accuracy samples for training, reiterating the need for scaleable, high-accuracy reference methods.

To enable DL-VMC methods to efficiently compute wavefunctions for many molecules, several methods have been proposed to amortize the cost of optimization, by learning a single wavefunction across multiple systems. This has been demonstrated to work for different geometries of a single molecule [10, 23, 24], and recently two approaches have been proposed to learn wavefunctions across entirely different molecules, each with their own limitations. Gao et al. [17] reparameterized the orbitals of a wavefunction, by using chemistry-inspired heuristics to determine orbital positions, managing to efficiently generalize wavefunctions across different geometries of a single molecule. However, when learning a single wavefunction across different molecules, their results deteriorated and transfer to new molecules proved difficult. Scherbela et al. [25] do not require heuristic orbital positions, but instead use orbital descriptors of a low-accuracy conventional method to parameterize DL-VMC orbitals. However, while their wavefunction ansatz successfully transfers to new molecules, their method requires a separate, iterative Hartree-Fock (HF) calculation for every new geometry.

**Our contribution**   This work presents the first end-to-end machine learning approach, which successfully learns a single wavefunction across many different molecules with high accuracy. Our contributions are:

- A transferable neural wavefunctions, which requires neither heuristic orbital positions, nor iterative HF calculations. We achieve this by building on the architecture by Scherbela et al. [25] and an orbital prediction model by Unke et al. [20].

- A simplified and improved electron embedding architecture, leveraging expressive nuclear features from our orbital prediction model and a message passing step between nuclei.

- A chemically diverse dataset with up to 100 molecules based on QM7-X [26], a data augmentation method based on normal mode distortions, and successful training of a neural wavefunction on these with continuously sampled geometries. Additionally, we improve the initialization of electrons around the molecule, reducing the computational overhead.

- As a final result, an accurate neural wavefunction, which shows for the first time zero-shot capabilities, i.e. high-accuracy energy predictions without additional optimization steps, on new systems. In particular it achieves better absolute energies than well established, high-accuracy gold-standard reference methods, such as CCSD(T)-3Z on unseen systems without any finetuning (cf. Fig. 2a).

**Overview of the paper** In Sec. 2 we outline our method and the procedure to optimize a transferable wavefunction across molecules. In Sec. 3 we thoroughly test the accuracy of our obtained wavefunction, by analyzing absolute energies (Sec. 3.1), relative energies (Sec. 3.2), and the impact of design choices in an ablation study (Sec. 3.3). Throughout this work, we compare against other high-accuracy methods, in particular results obtained by state-of-the-art DL-VMC methods and CCSD(T). Finally we analyze the scalability and limitations of our base model, by applying it to a large-scale dataset in Sec. 3.4, before a discussion and outlook for future research in Sec. 4.

## 2 Methods

Our approach is divided into two parts (cf. Fig. 1): On the one hand, a wavefunction ansatz, containing an electron embedding, orbital embedding and a Slater determinant. On the other hand a method for geometry sampling based on normal-mode distortions.

### 2.1 Our wavefunction ansatz

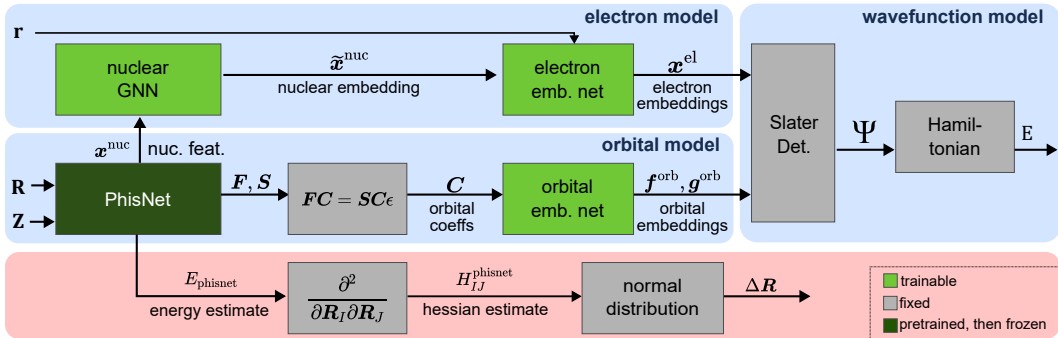

Figure 1: **Overview of our approach**: Wavefunction ansatz (top) and geometry sampling (bottom)

A single forward-pass for our wavefunction model

$$\psi = \sum_{d=1}^{N_{\text{det}}} \det\left[\phi_k^d(\boldsymbol{x}_i^{\text{el}})\right], \quad \phi_k^d(\boldsymbol{x}_i^{\text{el}}) = \sum_{I=1}^{N_{\text{nuc}}} \langle \boldsymbol{x}_i^{\text{el}}, \boldsymbol{f}_{Ikd}^{\text{orb}} \rangle e^{-r_{iI} g_{Ikd}^{\text{orb}}}, \quad i, k = 1, \ldots, n_{\text{el}} \tag{3}$$

can again be divided into three blocks: An electron model acting on nuclear and electron coordinates, generating electron embeddings $\boldsymbol{x}^{\text{el}}$; an SE(3)-equivariant orbital model acting only on nuclear coordinates, generating orbital embeddings $g^{\text{orb}}$ and $\boldsymbol{f}^{\text{orb}}$; a Slater determinant combining electron- and orbital-embeddings, and ensuring anti-symmetry of the wavefunction.

**Message passing neural network**    Throughout this work we use message passing neural networks (MPNN), to operate on the graph of particles which are connected by edges containing information about their relative positions. The electron-electron, electron-nuclear and nuclear-nuclear edges are embedded with a multi-layer perceptron (MLP),

$$\boldsymbol{e}_{ij}^{\text{el-el}} = \text{MLP}\big([\boldsymbol{r}_{ij}, r_{ij}]\big) \qquad \boldsymbol{e}_{iI}^{\text{el-nuc}} = \text{MLP}\big([\boldsymbol{r}_{iI}, r_{iI}]\big) \qquad \boldsymbol{e}_{IJ}^{\text{nuc-nuc}} = \text{MLP}\big([\boldsymbol{R}_{IJ}, R_{IJ}]\big), \quad (4)$$

by using a concatenation ($[\cdot]$) of distance and difference vectors, and separate weights for each MLP. A single message passing step is decomposed into the following operations

$$\tilde{\boldsymbol{a}}_i^{\text{rec}} = \text{MessagePassing}(\boldsymbol{a}_i^{\text{rec}}, \{\boldsymbol{a}_j^{\text{send}}\}, \{\boldsymbol{e}_{ij}\}) \tag{5}$$

$$= \sigma\bigg(\text{Linear}(\boldsymbol{a}_i^{\text{rec}}) + \text{Linear}\big(\sum_j \text{Linear}(\boldsymbol{a}_j^{\text{send}}) \odot \text{Linear}(\boldsymbol{e}_{ij})\big)\bigg) \tag{6}$$

for a receiving particle $\tilde{\boldsymbol{a}}_i^{\text{rec}}$ and the set of sending particles $\{\boldsymbol{a}_j^{\text{send}}\}$, connected via their edges $\{\boldsymbol{e}_{ij}\}$. By $\sigma$ we denote the non-linear activation and with $\odot$ the element-wise multiplication along the feature dimension. An MPNN is obtained by stacking MessagePassing layers

$$\text{MPNN}(\boldsymbol{a}_i, \{\boldsymbol{a}_j\}, \{\boldsymbol{e}_{ij}\}) = \text{MessagePassing}(\ldots \text{MessagePassing}(\boldsymbol{a}_i, \{\boldsymbol{a}_j\}, \{\boldsymbol{e}_{ij}\})) \tag{7}$$

In the following we use these message passing steps to model all inter-particle interactions.

**SE(3)-equivariant orbital model**    The orbital model is a simplified version of PhisNet[20], a neural network predicting the overlap matrix $\boldsymbol{S}$ and the Fock matrix $\boldsymbol{F}$, via nuclear embeddings $\boldsymbol{x}^{\text{nuc}}$:

$$\boldsymbol{x}^{\text{nuc}} = \text{phisnet}_\theta(\boldsymbol{R}, \boldsymbol{Z}) \qquad \boldsymbol{S}_{IJ} = s_\theta(\boldsymbol{x}_I^{\text{nuc}}, \boldsymbol{x}_J^{\text{nuc}}) \qquad \boldsymbol{F}_{IJ} = f_\theta(\boldsymbol{x}_I^{\text{nuc}}, \boldsymbol{x}_J^{\text{nuc}}). \tag{8}$$

Here $\boldsymbol{x}^{\text{nuc}} \in \mathbb{R}^{N_{\text{nuc}} \times (L+1)^2 \times N_{\text{channels}}}$, and $\boldsymbol{S}_{IJ}$ and $\boldsymbol{F}_{IJ}$ are each in $\mathbb{R}^{N_{\text{basis}} \times N_{\text{basis}}}$. The basis-set size of the predicted orbitals is denoted by $N_{\text{basis}}$ and the feature dimension of the nuclear embeddings by $N_{\text{channels}}$. The full overlap- and Fock-matrices are assembled from the corresponding blocks $\boldsymbol{S}_{IJ}$ and $\boldsymbol{F}_{IJ}$, leading to matrices of shape $[N_{\text{nuc}} N_{\text{basis}} \times N_{\text{nuc}} N_{\text{basis}}]$. Each layer of PhisNet is SE(3)-equivariant, ensuring that any 3D-rotation or inversion of the input coordinates $\boldsymbol{R}$, leads to an equivalent rotation of its outputs. This is done by splitting any feature vector into representations of varying harmonic degree $l = 0, \ldots, L$, each with components $m = -l, \ldots, l$. A detailed description of SE3-equivariant networks in general, as well as PhisNet in particular can be found in [20]. A list of changes and simplifications we made to PhisNet can be found in Appendix C.

The orbital embeddings (corresponding to orbital expansion coefficients in a conventional quantum chemistry calculation) are obtained by solving the generalized eigenvalue problem:

$$\boldsymbol{F}\boldsymbol{C}_k = \boldsymbol{S}\boldsymbol{C}_k\epsilon_k \qquad \hat{\boldsymbol{x}}_{Ik}^{\text{orb}} = \text{reshape}(\boldsymbol{C}_k, [N_{\text{nuc}}, N_{\text{basis}}])_I \tag{9}$$

It has been shown empirically that it is beneficial to obtain the orbital coefficients $\hat{\boldsymbol{x}}_{Ik}^{\text{orb}}$ as solutions to this generalized eigenvalue problem, rather than predicting them directly [27] as functions of $\boldsymbol{R}$ and $\boldsymbol{Z}$. This is because the orbital coefficients are neither unique, nor do they share the molecule's symmetry. The matrices $\boldsymbol{F}$ and $\boldsymbol{S}$ on the other hand are unique and transform equivariantly under E(3)-transformations of the molecule, leading to a well defined learning problem.

Following [25], we do not use the orbital energies $\epsilon_k$ and obtain the backflow factors $\boldsymbol{f}$ and exponents $\boldsymbol{g}$, by first localizing the resulting orbitals using the Foster-Boys localization [28] (cf. Appendix B) and subsequently using an MPNN and MLP acting on the orbital embeddings.

$$\widetilde{\boldsymbol{x}}_{Ik}^{\text{orb}} = \sum_{n=1}^{N_{\text{orb}}} U_{kn}^{\text{loc.}} \hat{\boldsymbol{x}}_{In}^{\text{orb}}, \qquad \boldsymbol{x}_{Ik}^{\text{orb}} = \text{MPNN}(\widetilde{\boldsymbol{x}}_{Ik}^{\text{orb}}, \{\widetilde{\boldsymbol{x}}_{Jk}^{\text{orb}}\}, \{\boldsymbol{e}_{IJ}^{\text{nuc-nuc}}\}) \tag{10}$$

$$\boldsymbol{f}_{Ik}^{\text{orb}} = \text{MLP}(\boldsymbol{x}_{Ik}^{\text{orb}}), \qquad \boldsymbol{f}_{Ik}^{\text{orb}} \in \mathbb{R}^{N_{\text{nuc}} \times N_{\text{orb}} \times N_{\text{det}} \times N_{\text{emb}}} \tag{11}$$

$$\boldsymbol{g}_{Ik}^{\text{orb}} = \text{MLP}(\boldsymbol{x}_{Ik}^{\text{orb}}), \qquad \boldsymbol{g}^{\text{orb}} \in \mathbb{R}^{N_{\text{nuc}} \times N_{\text{orb}} \times N_{\text{det}}} \tag{12}$$

**Electron model**    The electron embedding is a message passing neural network. To incorporate the geometric information of the molecule considered, we leverage the equivariant prediction of the PhisNet nuclear embeddings $\boldsymbol{x}^{\text{nuc}}$, by first performing a message passing step between the nuclear embeddings

$$\hat{\boldsymbol{x}}_I^{\text{nuc}} = \text{MLP}(\boldsymbol{x}_I^{\text{nuc}}) \qquad \tilde{\boldsymbol{x}}_I^{\text{nuc}} = \text{MessagePassing}(\hat{\boldsymbol{x}}_I^{\text{nuc}}, \{\hat{\boldsymbol{x}}_J^{\text{nuc}}\}, \{\boldsymbol{e}_{IJ}^{\text{nuc-nuc}}\})$$

and then using these features to initialize the electron embeddings

$$x_i^{\text{el},0} = \text{MessagePassing}(\mathbf{0}, \{\tilde{x}_J^{\text{nuc}}\}, \{e_{iJ}^{\text{el-nuc}}\}),$$

by using a zero vector $\mathbf{0}$ for the initial receiving electrons. This differentiates our electron model from previously proposed methods [3, 6, 8], leading to a better generalization when optimized across molecules (cf. Sec. 3.3). The final step of the embedding is a multi-iteration message passing between electron embeddings to capture the necessary electron-electron interaction

$$x_i^{\text{el}} = \text{MPNN}(x_i^{\text{el},0}, \{x_j^{\text{el},0}\}, \{e_{ij}^{\text{el-el}}\}),$$

resulting in a $N_{\text{emb}}$-dimensional representation for each electron $x_i^{\text{el}} \in \mathbb{R}^{N_{\text{emb}}}, i = 1, \ldots, n_{\text{el}}$.

**Overall E(3)-equivariance** Like existing approaches [3, 8, 17] our overall wavefunction does not enforce E(3)-symmetry. This is because the wavefunction can have lower symmetry than the molecule [23], for example in the case of the excited states of a hydrogen atom. We therfore choose all parts of the network that act purely on nuclear coordinates (i.e. the PhisNet model and the energy/hessian estimate) to be equivariant under E(3)-transformations. All parts of the network acting on electron coordinates (in particular the electron model) break this symmetry by depending explicitly on the cartesian coordinates of the electrons. The architecture is thus only invariant under translations, but not under rotations or inversions. We bias the model towards approximately invariant energies using data augmentation as discussed in Sec. 2.3.

## 2.2 Sampling

**Markov Chain Monte Carlo (MCMC) sampling of electron positions** We use MCMC to draw samples $r$ from the probability distribution $\psi(r)^2$, to evaluate the expectation value of Eq. 2. One notable difference compared to other works is our initialization $r^0$ of the Markov Chain. In the limit of infinite steps, the samples are distributed according to $\psi^2$, but for a finite number of steps the obtained samples strongly depend on $r^0$. This issue is typically addressed by a "burn-in", where MCMC is run for a fixed number of steps (without using the resulting samples) to ensure that $r$ has diffused to state of high probability. Previous work has initialized $r^0$ using a Gaussian distribution of the electrons around the nuclei. We find that this initialization is far from the desired distribution $\psi^2$ and thus requires $\approx 10^5$ MCMC steps to reach the equilibrium distribution. We instead initialize $r^0$ by samples drawn from an exponential distribution around the nuclei, which much better approximates the correct distribution and thus equilibrates substantially faster (cf. Appendix A). We find that exponential initialization reduces the required number of burn-in steps by ca. 50%, reducing the computational cost of a 500-step zero- shot evaluation by ca. 5%.

**Normal mode sampling of geometries** Since DL-VMC is an ab-initio method, we do not require a labeled dataset of reference energies, but to obtain a transferable wavefunction, which generalized well to new systems, a diverse dataset of molecular geometries $R$ is required. Starting with an initial set of geometries $R^0$, we update $R$ on the fly, by perturbing each geometry every 20 optimization steps by adding random noise $\Delta R$ to the nuclear coordinates. Using uncorrelated, isotropic random noise for $\Delta R$ would yield many non-physical geometries $R'$, since the stiffness of different degrees of freedom can vary by orders of magnitude. Intuitively we want to make large perturbations along directions in which the energy changes slowly, and vice-versa. We achieve this by sampling $\Delta R$ from a correlated normal distribution

$$\Delta R \sim \mathcal{N}(\alpha(R^0 - R), \beta H_{\text{phis}}^{-1}) \qquad R' = R + \Delta R. \tag{13}$$

The bias term $\alpha(R^0 - R)$ ensures that geometries stay sufficiently close to their starting point $R^0$. The covariance matrix is chosen proportional to the pseudo-inverse of the hessian of the energy $E^{\text{phis}}$, which is predicted from the scalar component of the nuclear embeddings using a pre-trained MLP. By using $H_{\text{phis}}^{-1}$ as covariance matrix, we take large steps along soft directions and small steps along stiff directions, thus avoiding unphysical geometries with very high energies.

$$E^{\text{phis}} = \sum_{I=1}^{N_{\text{nuc}}} \text{MLP}(x_I^{\text{nuc}}), \qquad H_{I\zeta,J\xi}^{\text{phis}} = \frac{\partial^2 E^{\text{phis}}}{\partial R_{I\zeta} \partial R_{J\xi}}, \quad I, J = 1 \ldots N_{\text{nuc}} \quad \zeta, \xi = 1 \ldots 3 \tag{14}$$

After distorting the nuclear coordinates $R$, we also adjust the electron positions $r$, using the space-warp coordinate transform described in [29], which effectively shifts the electrons by a weighted average of the shift of their neighbouring nuclei. In addition to this distortion of the molecule, we also apply a random global rotation to all coordinates, to obtain a more diverse dataset.

## 2.3 Optimization

To obtain orbital descriptors (and energies to calculate the hessian of the energy) we pre-train PhisNet against the Fock matrix, the overlap matrix, the energy and the forces of Hartree-Fock calculations in a minimal basis set across 47k molecules. Further details of the loss function, dataset, and the adaptions to PhisNet can be found in Appendix C. For all subsequent experiments, we freeze the parameters of PhisNet. A full DL-VMC calculation to obtain a ground-state energy prediction can be divided into three consecutive steps:

1. **Supervised optimization using PhisNet**: Initially, the neural-network orbitals (cf. Eq. 3) are optimized to minimize the residual against orbitals obtained from PhisNet. It ensures that the initial wavefunction roughly resembles the true ground-state and is omitted when fine-tuning an already optimized base model.

2. **Variational optimization**: Minimization of the energy (cf. Eq. 2) by drawing samples from $\psi^2$ using MCMC and updating the wavefunction parameters using the KFAC optimizer [30].

3. **Evaluation**: For inference of the ground-state energy, we sample electron positions using MCMC, and evaluate the energy using Eq. 2 without updating $\theta$.

To train a multi-geometry transferable wavefunction we further divide the variational optimization of a neural wavefunction into two steps:

1. **Pre-training**: A single wavefunction model is trained across many molecules and geometries. In every gradient step we only consider a single geometry per batch. The next geometry to optimize is chosen based on the energy variance as proposed by [10]. To sample continuously the space of molecular geometries we distort each geometry every 20 optimization steps as described in Sec. 2.2. We refer to evaluations of this model on new systems as "zero-shot".

2. **Fine-tuning**: A a small number of additional variational optimization steps is done using geometries of interest, starting from the weights of a pre-trained base model. This procedure yields a model that is specialized to the molecule at hand and typically yields more accurate energies on the specific problem than the raw pre-trained model.

## 3 Results

We pre-train our wavefunction model on a dataset of 98 molecules (699 conformers) for 256k optimization steps using the architecture and training procedure outlined in Sec. 2. Below we demonstrate the performance of this model, for zero-shot evaluations and after subsequent fine-tuning.

### 3.1 Accuracy of pre-trained model for absolute energies

To test the transfer capabilities of the model, we evaluate it on test-sets, which each contain 4 randomly chosen and perturbed molecules, grouped by molecule size (measured as the number of non-Hydrogen atoms). To avoid train/test leakage, we excluded all molecules that are part of these test sets from the training set (cf. Appendix D). Although the model has only been trained on molecules containing up to 4 heavy atoms, we evaluate its performance across the full range up to 7 heavy atoms. We find that for molecules containing up to 6 heavy atoms our method outperforms CCSD(T) with a 2Z basis set and outperforms CCSD(T) with a 4Z basis set after only 4k fine-tuning steps. This is a large improvement over the state of the art: Zero-shot evaluations by Gao et al. [17] did not manage to outperform a Hartree-Fock baseline, even on the toy system of Hydrogen-chains. Similarly, Scherbela et al. [25] achieve high accuracy after fine-tuning, but result that are worse than Hartree-Fock in a zero-shot setting. In contrast, our improvements to their method increase zero-shot accuracy by more than 2 orders of magnitude.

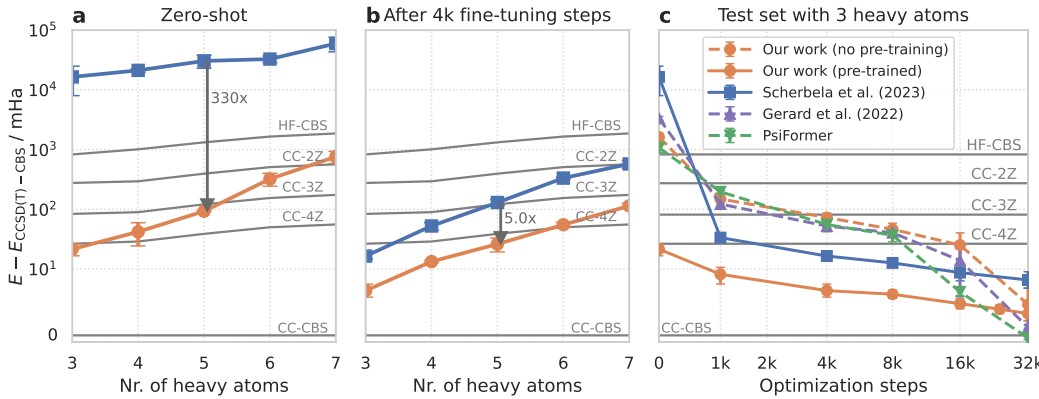

Figure 2: **Absolute energies**: Energies relative to CCSD(T)-CBS (complete basis set limit) when re-using the pre-trained model on molecules of varying size without optimization (a) and after fine-tuning (b). (c) depicts energy for the test set containing 3 heavy atoms as a function of optimization steps and compares against SOTA methods. Solid lines are with pre-training, dashed lines without. Gray lines correspond to conventional methods: Hartree-Fock in the complete basis set limit (HF-CBS), and CCSD(T) with correlation consistent basis basis sets of double to quadruple valence (CC-nZ).

We furthermore find that fine-tuning pre-trained wavefunctions can be more cost effective than well established conventional methods. For a typical molecule containing 5 heavy atoms, we require 9.5 node-hours for 4k fine-tuning steps, achieving accuracy that surpasse CCSD(T)-4Z. In contrast, CCSD(T)-4Z requires 10.5 node-hours and 16k steps of PsiFormer (achieving the same accuracy), require 53 node-hours. All run-times are listed in Tab. 3 of Appendix I.

To further test the accuracy of our method, we evaluate for 3 heavy atoms the performance with increasing fine-tuning steps (cf. Fig. 2c). We compare our work (with and without fine-tuning) against CCSD(T) and reference calculations done with state-of-the-art DL-VMC methods [6, 8]. For up to 16k optimization steps, our pre-trained model yields energy errors that are 1-2 orders of magnitude lower than other reference methods. After longer optimization the method by Gerard et al. [8] and PsiFormer [6] surpass our predictions. We hypothesize that this is not to blame on pre-training, but that our orbital prediction framework, which allows us to optimize across molecule, yields less expressive wavefunctions than a fully trainable backflow. This is demonstrated by the fact that a pre-trained and non-pre-trained model converge to the same energy in the limit of long optimization.

Like the absolute energies, also the variance of the local energies (another measure of wavefunction accuracy) is substantially improved by our method and results are depicted in Appendix E.

### 3.2 Accuracy for relative energies

While Sec. 3.1 demonstrates high accuracy for absolute energies, we find that relative energies (being the small difference between two large absolute energies) can be unsatisfactory in the zero-shot regime, and a small number of fine-tuning steps is required to reach quantitatively correct relative energies. Fig. 3 demonstrates this issue on 4 distinct systems, each highlighting a different challenge for our model. For each system, we evaluate our pre-trained base model without any system specific optimization (zero-shot), and after 4000 fine-tuning steps. The fine-tuning optimization is done separately for each of the 4 systems, analogously to Sec. 2.3, yielding 4 distinct wavefunctions that each represent the ground-state wavefunctions of all considered geometries per system. Results after more fine-tuning steps can be found in Appendix F.

**Bicyclobutane conformers** Fig. 3a depicts the energies of 5 conformers of bicyclobutane relative to the energy of its initial structure. The system is of interest, because CCSD(T) severely underestimates the energy of the dis-TS conformer by $\approx 60$ mHa [31]. While our zero-shot results yield the correct sign for the relative energies, they are quantitatively far off from the gold-standard DMC reference calculation [31], in particular for the dis-TS geometry, where we overestimate the relative energy

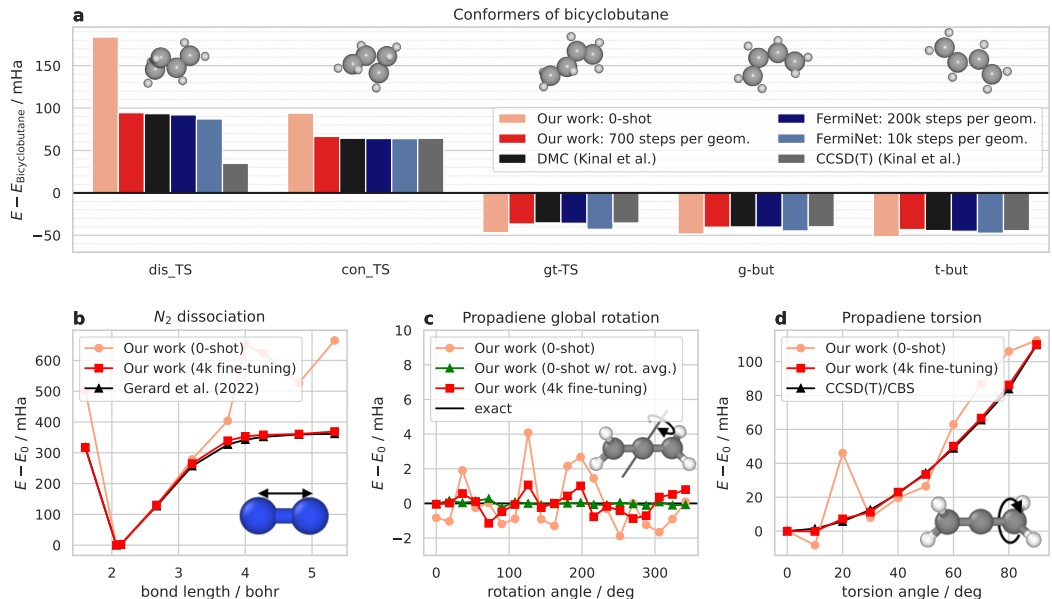

Figure 3: **Challenging relative energies**: Relative energies obtained with and without fine-tuning on 4 distinct, challenging systems, compared against high-accuracy reference methods. a) Relative energy of bicyclobutane conformers vs. the energy of bicyclobutane b) Potential energy surface (PES) of $N_2$, c) global rotation of propadiene d) relative energy of twisted vs. untwisted propadiene.

by 90 mHa. However when fine-tuning our model for only 700 steps per geometry (4k total), we obtain relative energies that are in close agreement with DMC (max. deviation 2.1 mHa). Comparing to FermiNet [5] we find that our results are more accurate than a FermiNet calculation after 10k steps (requiring twice our batch size; max. deviation 7.5 mHa), and slightly less accurate than a FermiNet calculation optimized for 200k steps (max. deviation 1.4 mHa). As opposed to CCSD(T) our model does not suffer from systematic errors, even for the challenging dis-TS geometry. A table of all relative energies can be found in Appendix F.

**Nitrogen dissociation**    When evaluating the energy of an $N_2$ molecule at various bond-lengths we obtain high zero-shot accuracy near the equilibrium geometry ($d = 2.1$ bohr; in training set), but substantially lower accuracy at smaller or large bond lengths (cf. Fig. 3b). This is due to the lack of dissociated atoms in the pre-training dataset and again mostly remedied by finetuning. In the most challenging regime around $d = 3.7$, even with fine-tuning we obtain energies that are 12 mHa above high-accuracy results obtained by Gerard et al. [8], indicating the need for longer fine-tuning.

**Global rotation of propadiene**    Energies of molecules are invariant under global translation or rotation of all particle positions. The wavefunction however is not invariant and neither is our wavefunction ansatz, which can lead to different energies for rotated copies of a molecule. When evaluating the energy of our model on 20 copies of propadiene ($C_3H_4$) rotated around a random axis, we find typical energy variations of $\pm 1$ mHa, but also a individual outliers, deviating by up to 5 mHa. This highlights a dilemma facing all existing DL-VMC models: On the one hand, constraining the wavefunctions to be fully invariant under rotation (or even just invariant under symmetries of the Hartree-Fock orbitals) is too restrictive to express arbitrary ground-state wavefunctions [23]. One the other hand, our approach of biasing the model towards rotation-invariant energies by data augmentation, appears to be helpful but not to be fully sufficient. When evaluating an earlier checkpoint of our base model (trained for 170k epochs instead of 256k), we observe mean energy variations of $\pm 3$ mHa and outliers of up to 30 mHa, indicating the positive impact of prolonged training with data augmentation. We again find 4000 fine-tuning steps split across all geometries are sufficient to reduce errors below chemical accuracy of 1.6 mHa.

In addition to augmenting data during training time, we can also augment the data during inference time, to obtain fully rotation-invariant energies, from an approximately rotation invariant model. We

achieve this by randomly rotating the molecule at every inference step and averaging across all rotated geometries. Since the Monte-Carlo estimate is already an average across many different samples anyways, this comes at no additional computational cost. Fig. 3c shows that with rotation averaging we obtain energies that are fully invariant, up to Monte-Carlo noise.

**Twisted propadiene** Twisting one of the C=C bonds of propadiene leads to a transition state with an energy difference of 110 mHa. Evaluating the energy without fine-tuning on an equidistant grid of torsion angles, we obtain the correct barrier height (112 mHa), but also deviations of up to 40 mHa. During pre-training we purposefully sampled twisted molecules, but only included equilibrium geometries, transition geometries, and one intermediate twist (cf. Appendix D). This seems sufficient for correct zero-shot barrier heights, but insufficient for high accuracy along the full path. Short fine-tuning (4k steps distributed across all 10 geometries) yields excellent agreement with CCSD(T): ~0.2 mHa discrepancy for the barrier height and a maximum deviation of 2 mHa along the path.

### 3.3 Ablation studies

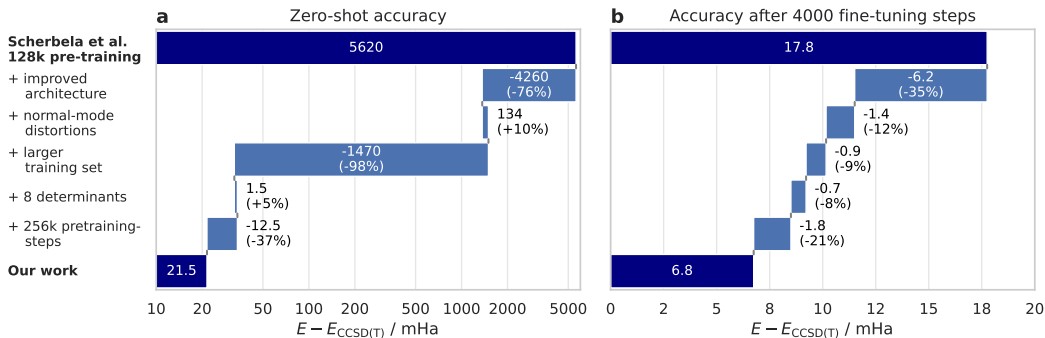

Figure 4: **Ablation study**: Breakdown of absolute energy difference between Scherbela et al. [25] and our work, when evaluating the respective base models on the test set consisting of 3 heavy atoms. Panel a shows accuracy gains for zero-shot evaluation, panel b the accuracy after 4k fine-tuning steps. First and last row depict their and our absolute energy respectively, intermediate bars show the subsequent improvements of various changes in mHa (and %). Note that panel a. is plotted on a logarithmic scale due to the large energy difference.

To analyze the relative importance of our changes, we break down the accuracy gap between our work and the prior work by Scherbela et al. [25] in Fig. 4. We start with their model checkpoint trained for 128k epochs on their dataset consisting of 18 different compounds. Next, we train a model using our improved architecture with their dataset and methodology, already reducing the zero-shot error by 76% and fine-tuning error by 35%. The next model additionally uses normal-mode distortions to augment the training dataset, decreasing the fine-tuning error by another 12%. Additionally increasing the training set size to our dataset containing 98 molecules and the corresponding torsional conformers, reduces the zero-shot error by 98% and yields modest improvements in the fine-tuning case. Increasing the number of determinants from 4 to 8 and increasing the number of pre-training steps from 128k to 256k improve fine-tuning accuracy by 8% and 21% respectively. The two largest contributions overall are the improved architecture – yielding substantial gains both in zero-shot and fine-tuning – as well as the larger training set, which is crucial for high zero-shot accuracy. This is consistent with [25], which found that both model size and training set size do improve performance, while pre-training duration shows diminishing returns after 256k steps. Since we pre-train the PhisNet model against Hartree-Fock references and do not update its parameters during optimization, the PhisNet model by itself contributes no substantial accuracy improvement. It impacts energy predictions indirectly, by providing pre-trained nuclear emebddings $x^{\mathrm{nuc}}$ and enabling fast geometry distortions and rotations during training and inference, which in turn improve accuracy.

### 3.4 Large-scale experiment

To showcase the scalability of our approach, we evaluate zero-shot predictions of the absolute energies for a subset of 250 molecules from the QM7 dataset [32], containing molecules with 14-58 electrons.

Since CCSD(T) calculations would be prohibitively expensive for a dataset of this size, in Fig. 5a we compare against density functional theory (DFT) reference calculations (PBE0+MBD, [26]). Fig. 5b breaks down the energy differences compared to DFT, by molecule size. For molecules with less than 5 heavy atoms we obtain energies that are lower than DFT by 10-120 mHa. Because our ansatz is variational (in contrast to DFT), our lower energies translate to a more accurate prediction of the true ground-state energy, confirming again our high zero-shot accuracy. With increasing system size the accuracy deteriorates, due to increasing extrapolation and out-of-distribution predictions, consistent with our results in Sec. 3.1. One reason for the large energy differences in larger molecules is the increasing inter-atomic distance within the molecules with increasing number of atoms (cf. Fig. 5c). While the largest inter-atomic distance observed in the training set is 11 bohr, the evaluation set contains distances up to 17 bohr. This issue could be overcome by employing a distance cutoff, as it is already applied in supervised machine-learned potential energy surface predictions [19, 33] or has just recently been incorporated into a neural wavefunction [17] via an exponential decay with increasing inter-particle distance. Another potential solution is to include larger molecules or separated molecule fragments in the pre-training molecule dataset, reducing the extrapolation regime.

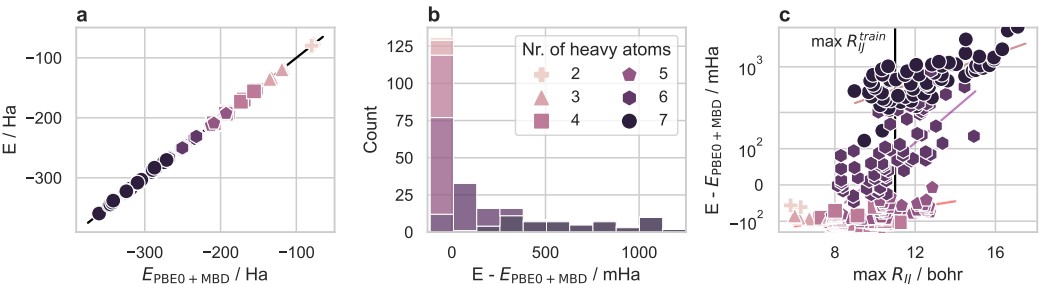

Figure 5: **Zero-shot on QM7**: a) Our zero-shot energies vs. DFT [26] b) Histogram of energy residuals (truncated at 1.25 Ha for clarity) c) Residuals vs. largest inter-molecular distance $R_{IJ}$.

## 4 Discussion

We have presented to our knowledge the first ab-initio wavefunction model, which achieves high-accuracy zero-shot energies on new systems (Sec. 3.1 and Sec. 3.4). Our pre-trained wavefunction yields more accurate total energies than CCSD(T)-2Z across all molecule sizes and outperforms CCSD(T)-3Z on molecules containing up to 5 heavy atoms, despite having been trained only on molecules containing up to 4 heavy atoms. We find that relative energies of our model are qualitatively correct without fine-tuning, but need on the order of 4000 fine-tuning steps to reach chemical accuracy of 1.6 mHa (Sec. 3.2). This is a substantial improvement over previous work, which so far has fallen in two categories: High-accuracy ansätze (such as [3, 6]) that cannot generalize across molecules and thus need ca. 10x more compute to reach the same accuracy, or methods that can generalize ([17, 25]), but yield orders of magnitude lower accuracy in the zero- or few-shot regime. We demonstrate in Sec. 3.3 that these improvements are primarily driven by an improved architecture (containing a more expressive electron embedding and an ML-based orbital model), improved geometry sampling, and a larger training dataset.

While results are encouraging in the zero- and few-shot regime, open questions for further research abound. The most pressing issues are currently limited zero-shot accuracy for relative energies, and potentially limited expressiveness of the ansatz in the regime of very long optimization. Zero-shot accuracy could be further improved by training on an even larger dataset, further improved geometry sampling (in particular of torsion angles), and an interaction cut-off to avoid previously unseen particle pairs for new large molecules. Furthermore SE(3)-symmetry of the wavefunction should be explored further, since currently only the orbital part of our architecture is SE(3)-equivariant. We experimented with a fully equivariant architecture, but found the resulting wavefunctions to not be expressive enough. To improve overall accuracy, attention based embeddings [6, 34] could be pursued. Additionally, we currently freeze the weights of the orbital embedding to simplify the architecture and avoid back-propagation through the iterative orbital localization procedure. Optimizing these weights in addition to the electron embedding will lead to a more expressive ansatz.

## Acknowledgements

We gratefully acknowledge financial support from the following grants: Austrian Science Fund FWF Project I 3403 (P.G.), WWTF-ICT19-041 (L.G.). The computational results have been achieved using the Vienna Scientific Cluster (VSC). The funders had no role in study design, data collection and analysis, decision to publish or preparation of the manuscript.

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

# Supplementary material for
# Variational Monte Carlo on a Budget –
# Fine-tuning pre-trained Neural Wavefunction

## A   Electron MCMC initialization

To investigate the impact of the initial distribution of electron positions on the equilibration of the Markov Chain, we run two evaluations for a glycine molecule, using a pre-trained wavefunction. We perform no initial burn-in and use every 50th sample for energy evaluation. If the chain was perfectly equilibrated right after initialization, all sampled energies would fluctuate around the mean energy. However as Fig. 6b shows, it takes several thousand steps for the sampled energies to converge to the correct mean. This is particularly pronounced with Gaussian initialization of electron positions, which is the default in state-of-the-art DL-VMC codes such as FermiNet [3]. Using an exponential distribution of the initial electron positions much more closely resembles the correct electron density $\psi^2$ (cf. Fig. 6a) and thus reaches equilibrium substantially faster.

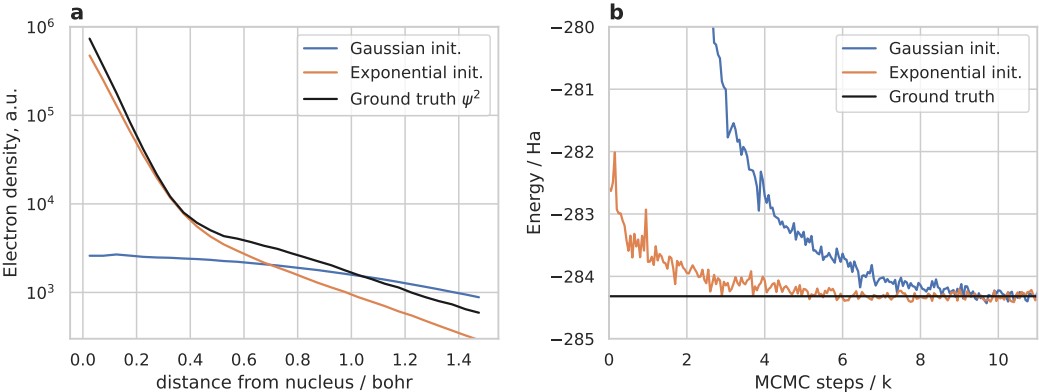

Figure 6: **Effect of electron initialization**: Initializing the electron positions using an exponential distribution instead of Gaussian, better fits the actual density (a), and thus leads to faster equilibration of observables during evaluation (b).

## B   Orbital localization

Our model uses orbital embeddings $\boldsymbol{x}^{\text{orb}}$ as inputs to parameterize the backflows $\boldsymbol{f}^{\text{orb}}$, and exponents $\boldsymbol{g}^{\text{orb}}$ of the orbitals. These orbital embeddings were introduced by Scherbela et al. [25] in the form of molecular orbital expansion coefficients, obtained from a self consistent Hartree-Fock calculation. In this setting, the coefficients $\boldsymbol{x}^{\text{orb}}$ are not uniquely defined, but only up to a linear transformation $U$ with determinant $\pm 1$

$$\boldsymbol{x}^{\text{orb}}_{Ik} = \sum_{n=1}^{N_{\text{orb}}} U_{kn} \hat{\boldsymbol{x}}^{\text{orb}}_{Ik}, \qquad U \in \mathbb{R}^{N_{\text{orb}} \times N_{\text{orb}}}, \qquad \det U = \pm 1. \tag{15}$$

This stems from the fact that the corresponding Hartree-Fock wavefunction is invariant under such a transformation. Consequently there is free choice, which linear combination of embeddings $\boldsymbol{x}^{\text{orb}}$ to choose from without any loss of information. We follow the approach of [25], by choosing $U$ such that the corresponding Hartree-Fock orbitals are maximally localized according to the Foster-Boys

metric, i.e. minimize the spatial variance $\mathcal{L}$:

$$\phi_k(\boldsymbol{r}, U) = \sum_{I=1}^{N_{\text{nuc}}} \sum_{\mu=1}^{N_{\text{basis}}} b_{I\mu}(\boldsymbol{r}) U_{kn} \hat{x}_{In\mu}^{\text{orb}} \tag{16}$$

$$\mathcal{L}(U) = \sum_k \int \phi_k^2(\boldsymbol{r}, U) \boldsymbol{r}^2 d\boldsymbol{r} - \left( \int \phi_k^2(\boldsymbol{r}, U) \boldsymbol{r} d\boldsymbol{r} \right)^2 \tag{17}$$

Here $b_{I\mu}(\boldsymbol{r})$ denotes $\mu$-th basis function of the Hartree-Fock expansion, centered on the $I$-th nucleus. In practice the integrals of Eq. 17 do not have to be evaluated explicitly, but can instead be computed via the overlap matrix $\boldsymbol{S}$. The minimization of $\mathcal{L}$ is typically done iteratively, requires on the order of 10 steps, and is readily implemented in many open-source quantum chemistry codes such as pySCF [35].

## C  Adaption of PhisNet

We heavily rely on PhisNet by Unke et al. [20] to obtain orbital descriptors without the need for a separate SCF calculation. Compared to their original work, we made several simplifications, which are motivated by the fact that we do not predict final high-accuracy orbitals in a large basis set, but only use PhisNet as a feature extractor by predicting orbitals in a minimal basis-set:

- **Layer Norm** We found deep variants of PhisNet to be unstable to train and mitigated the issue by adding an (equivariant) layer norm after each PhisNet module.

- **Simplified Fock matrix prediction** The original PhisNet implementation uses a final interaction between the node embeddings, before predicting the elements of the Fock matrix. We found this interaction to be superfluous for our purposes and left it out for simplicity.

- **Separate energy head** The original PhisNet computes energies via the eigenvalues obtained by diagonalization of the Fock matrix. We instead predict energies using a separate head on top of the scalar features of the node embeddings.

- **Smaller network** We changed the hyperparameters to obtain a smaller and faster version of PhisNet which obtained sufficient accuracy for our purposes. We used 2 layers (instead of 5) and $L_{\text{max}} = 2$ (instead of 4). This reduces the number of parameters from 17M to 3M.

- **Diverse training set** While the original work optimized separate models for each molecule (e.g. by training on different geometries of a molecular dynamics simulation), we optimize a single model to predict $\boldsymbol{F}$, $\boldsymbol{S}$, $\boldsymbol{E}$, and $\boldsymbol{\nabla}\boldsymbol{E}$ across a dataset of 47k geometries sampled from QM7-X [26].

- **JAX re-implementation** We re-implemented PhisNet in JAX, using the e3nn library [36] to construct the SE(3)-equivariant operations.

We train the PhisNet-model on a dataset of 47k molecules from QM7X [26], using the Adam optimizer [37] on the following loss

$$\mathcal{L} = \sum_n \left( E^{\text{phis}}(\boldsymbol{R}^n, \boldsymbol{Z}^n) - E^{\text{ref},n} \right)^2 + \tag{18}$$

$$+ \sum_{nI\zeta} \left( \frac{\partial}{\partial R_{I\zeta}^n} E^{\text{phis}}(\boldsymbol{R}^n, \boldsymbol{Z}^n) - G_{I\zeta}^{\text{ref},n} \right)^2 + \tag{19}$$

$$+ \sum_{nIJ\mu\nu} \left( F_{IJ\mu\nu}^{\text{phis}}(\boldsymbol{R}^n, \boldsymbol{Z}^n) - F_{IJ\mu\nu}^{\text{ref},n} \right)^2 + \tag{20}$$

$$+ \sum_{nIJ\mu\nu} \left( S_{IJ\mu\nu}^{\text{phis}}(\boldsymbol{R}^n, \boldsymbol{Z}^n) - S_{IJ\mu\nu}^{\text{ref},n} \right)^2 . \tag{21}$$

Here $E$ denotes energies, $G$ denotes gradients of energies, $F$ Fock matrices, and $S$ overlap matrices. The indices $I, J$ run over nuclei, the indices $\mu, \nu$ over basis functions, and the index $n$ over samples in a batch.

## D  Molecule datasets

**Bicyclobutane**  For the Bicyclobutane to 1,3-butadiene transition we use the geometries from Kinal et al. [31] and compare against the reference energies stated in Spencer et al. [5].

**N$_2$**  For the N$_2$ potential energy surface with various bond-lengths we used the geometries including reference calculations from [8].

**Propadiene**  The global rotation of 360° degrees for propadiene is performed on the geometry which is part of the test set for 3 heavy atoms. For the torsion experiment we used the equilibrium geometry and rotated the torsion angle by 90° degrees in steps of 10° degrees.

**Zero-shot and fine-tuning dataset**  The results on zero-shot and few-shot predictions for increasing number of heavy atoms are performed on random subsets of molecules. For 5-7 heavy atoms we sample 4 unique and distorted molecules from QM7-X [26]. For 4 heavy atoms we use all geometries from the Bicyclobutane dataset. For 3 heavy atoms we use the ablation dataset.

**Ablation dataset**  For the ablation study, we use one geometry per molecule from the out-of-distribution test set from Scherbela et al. [25], leading to a set of four distinct molecules. We ensure that these molecules are not part of the training set.

**Large scale experiment**  For the large scale experiment we used a stratified random sample of 250 molecules from QM7 [32]. It contains all molecules with up to 4 heavy atoms, and additionally 65 randomly chosen molecules for 5, 6 and 7 heavy atoms each.

**Pre-training dataset for transferable neural wavefunctions**  To train our pre-trained wavefunctions we use two datasets, consisting of 18 and 98 disparate molecules. For part of the ablation we use the dataset proposed in [25] and an extended version with 80 additional molecules. The additional compounds are a combination of all valid SMILES generated with RDKit [38] with 3 heavy atoms, allowing only Nitrogen, Oxygen and Carbon with single-, double- or triple-bonds, and all molecules up to four heavy atoms from QM7-X [26] (excluding molecule containing Fluorine). To prevent a train-test leakage, we remove Bicyclobutane (including all conformations) and the four molecules from the ablations dataset. Since the normal-mode-distortions by design do not generate strongly distorted geometries, we augment the 98-molecule-dataset with rotated dihedral angles. To generate a subset of all possible dihedral angles for a heavy-atom bond we first generate samples with equidistant angles for all possible dihedral angles and compute Hartree-Fock energies with a minimal basis-set. We include the equilibrium geometry and all extrema of the potential energy surface with respect to the rotation of a single dihedral angle if the energy of the extrema is significantly different to already included geometries of the same molecule. Additionally, we include the transition geometry towards the respective extrema and again only include energetic diverse states. Finally, to make sure that certain molecules are not underrepresented in the dataset we make sure that all molecules have at least 5 geometries that get distorted during pre-training by adding copies of the equilibrium geometry. Overall this yields 699 initial geometries $R^0$ for pre-training.

## E  Energy variance

Fig. 7 depicts the energy variance obtained by various models. Analogously to the the energies presented in Fig. 2 of the main text, we also find that our pre-trained model achieves substantially lower energy variance compared to previous generalizing wavefunctions. Notably on the test set with 3 heavy atoms, the variance of our zero-shot model is on par with the variance of a PsiFormer model trained for 16k steps.

## F  Relative energies

To better depict the accuracy of the models' relative energies for the test systems in Fig. 3 of the main text, we depict the difference of relative energies between our model and a references method in Fig. 8. Furthermore we list numerical values vor all energies of Fig. 3a in Tab. 1. We find that on these

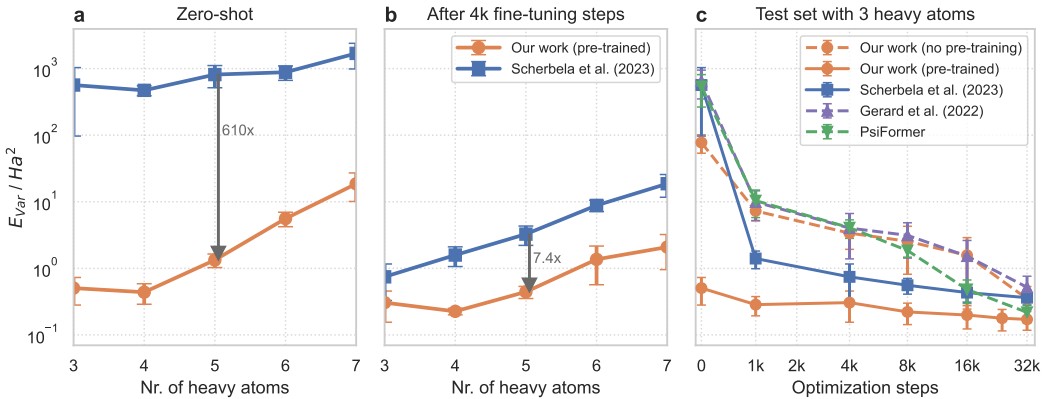

Figure 7: **Variance of energies**: Variance when re-using the pre-trained model on molecules of varying size without optimization (a) and after fine- tuning (b). For the test set containing 3 heavy atoms as a function of optimization steps, comparing against current state-of-the-art methods. Solid lines correspond to pre-trained models, dashed lines to models without pre-training (c).

challenging systems our model yields relative energy errors of up to 15 mHa when only fine-tuned for 400 steps per geometry. When fine-tuning our pre-trained model for 3200 steps per geometry, we obtain relative energies that are accurate within 2-3 mHa.

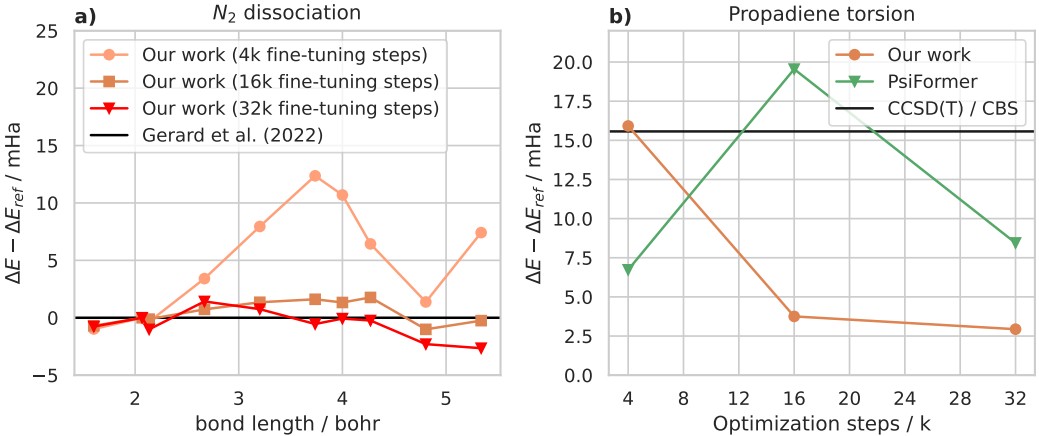

Figure 8: **Challenging relative energies**: The difference between relative energies of a reference method vs our work for two challenging systems. **a) Nitrogen dimer:** Difference of relative energies to the equilibrium geometry for our approach to reference calculation from Gerard et al. for $N_2$, **a) Propadiene:** Difference of relative energies of the transition barrier of twisted vs. untwisted propadiene. As reference method we choose PsiFormer with 64k optimization steps. This figure complements Fig. 3b,d of the main text with results for additional fine-tuning steps.

## G  Reference energies

**CCSD(T)**    All CCSD(T) energies – except explicitly stated otherwise – were obtained using ORCA [39] starting from a restricted Hartree-Fock calculation. We use correlation consistent basis sets of the cc-pCVXZ family, with X in {2, 3, 4}. To extrapolate to the complete basis set limit (CBS), we use

| structure | CCSD(T) [31] | DMC [31] | FermiNet 200k [5] | FermiNet 10k [5] | Our work zero-shot | Our work 700 per geom |
|---|---|---|---|---|---|---|
| con_TS | 64.4 | 64.4 | 64.1 | 63.9 | 94.0 | 66.6 |
| dis_TS | 34.7 | 93.4 | 92.0 | 87.1 | 183.8 | 94.5 |
| g-but | -40.0 | -40.2 | -40.3 | -44.9 | -48.5 | -40.4 |
| gt-TS | -35.5 | -35.4 | -35.9 | -42.9 | -46.9 | -36.7 |
| t-but | -44.6 | -44.5 | -45.3 | -47.5 | -51.8 | -43.2 |

Table 1: Energies relative to the energy of bicyclobutane in mHa, including the zero-point vibrational energy correction from Kinal et al. [31]. This data complements Fig. 3a in the main text.

the approach outlined in [3] and fit the following functions with free parameters $E_{\text{CBS}}^{\text{HF}}, E_{\text{CBS}}^{\text{corr}}, a, b, c$:

$$E_X^{\text{HF}} = E_{\text{CBS}}^{\text{HF}} + ae^{-bX}$$
$$E_X^{\text{corr}} := E_X^{\text{HF}} - E_X^{\text{CCSD(T)}} = E_{\text{CBS}}^{\text{corr}} + cX^{-3}$$
$$E_{\text{CBS}}^{\text{CCSD(T)}} = E_{\text{CBS}}^{\text{HF}} + E_{\text{CBS}}^{\text{corr}}$$

We stress that although CCSD(T)-energies are often considered as "gold-standard", they do not necessarily represent the actual ground-state energy. There are many cases, where CCSD(T) either overestimates the true ground-state energy, or even underestimates it, because CCSD(T) does not yield upper bounds to the true ground-state energy.

**PsiFormer**   For Fig. 2 we used the open-source FermiNet codebase [40]. The codebase didn't allow for inference calculation, therefore a slight fix was applied. All calculations were performed with the small settings as proposed in von Glehn et al. [6].

# H   Hyperparameters

A detailed description of the hyperparameter used in this work can be found below (cf. Tab. 2). For the mapping of the orbital descriptors to the electron embeddings to build the orbitals we rely on the hyperparameter from [25]. For optimization we rely on the second-order method KFAC [30] and use their Python implementation [41]. During the continuous sampling of the geometries we allow each geometry to perform a maximum of 20 steps of normal-mode distortion from the initial geometry and reset to the original one once the threshold is reached.

# I   Runtime and computational resources

Tab. 3 lists the run-times for our method and a comparable conventional reference method (CCSD(T)-4Z) for a typical molecule with 5 heavy atoms. We find that our fine-tuning our model for 4k steps yields lower absolute energies than CCSD(T)-4Z (cf. Fig. 2) at similar computational cost. This is in contrast to many earlier works that achieve highly accurate energies, but at computational cost that far surpasses the cost of conventional methods for small molecules.

Overall we used $\approx 5$k GPUhs (A100) for development and training of our base models, and another 5k GPUhs (A40) on evaluations and fine-tuning. Additionally we required $\approx 20$k CPUhs for CCSD(T) reference calculations.

| Method | Runtime / h |
|---|---|
| Our work, zero-shot | 1.0 |
| Our work, 4k fine-tuning | 9.5 |
| PsiFormer, 16k steps | 53.0 |
| CCSD(T), 4Z basis | 10.5 |

Table 3: **Runtime** of various quantum chemistry methods for $C_4OH_6$. All timings are in node hours, being 2 A100-hours for our work and PsiFormer, and 256 CPU-hours for CCSD(T).

## J Code and data availability

All code, configuration files, geometries, datasets and obtained energies are available on GitHub under `https://github.com/mdsunivie/deeperwin`. Model weights are available on figshare under `https://doi.org/10.6084/m9.figshare.23585358.v1`.

| | | |
|---|---|---|
| **Electron Embedding** | Hidden dimension $N_{\text{emb}}$ | 256 |
| | № iterations | 4 |
| **Nuclear Embedding** | Hidden dimension $\tilde{x}^{nuc}$ | 64 |
| | № layer MLP | 1 |
| **Message passing** | Activation function | SiLU |
| | № layer edge embedding | 3 |
| | Dimension edge embedding | 64 |
| | Dimension linear layer | 32 |
| **Markov Chain Monte Carlo** | № walkers | 2048 |
| | № decorrelation steps | 50 |
| | Target acceptance prob. | 50% |
| **PhisNet [20]** | Pre-trained against basis set | STO-6G |
| | № iterations | 2 |
| | Harmonic degree L | 2 |
| | № radial basis functions | 128 |
| | Hidden dimension of $x^{nuc}$ | 128 |
| | Distance cutoff (bohr) | 30 |
| **Transferable atomic orbitals [25]** | № determinants $N_{\text{det}}$ | 8 |
| | № hidden layers $f^{\text{orb}}$ | 2 |
| | Hidden dimension of $f^{\text{orb}}$ | 256 |
| | № hidden layers $g^{\text{orb}}$ | 2 |
| | Hidden dimension $g^{\text{orb}}$ | 128 |
| | № iterations MPNN | 2 |
| | № radial basis functions | 16 |
| | Hidden edge embedding dimension | 32 |
| | Hidden node embedding dimension | 16 |
| | Activation function | SiLU |
| **Variational pre-training** | Optimizer | KFAC |
| | Batch size | 2048 |
| | Norm constraint | $3 \times 10^{-3}$ |
| | Initial damping $d_0$ | 1 |
| | Minimal damping $d_{\text{min}}$ | 0.001 |
| | Damping rate decay | $d(t) = d_0 \exp(-t/20000)$ |
| | Initial learning rate $lr_0$ | 0.1 |
| | Learning rate decay | $lr(t) = lr_0(1 + t/6000)^{-1}$ |
| | Optimization steps | 128,000 - 256,000 |
| **Changes for fine-tuning** | Learning rate decay | $lr(t) = lr_0(7 + t/6000)^{-1}$ |
| | Optimization steps | 0 - 32,000 |
| **Sampling geometries** | Distortion energy $\beta$ | 0.005 Ha |
| | Max age | 20 |
| | Bias towards original geometry $\alpha$ | 0.2 |

Table 2: Hyperparameter settings used in this work

