# OpenReview forum: "Variational Monte Carlo on a Budget — Fine-tuning pre-trained Neural Wavefunctions"
_NeurIPS.cc/2023/Conference — NeurIPS 2023 poster_

### Official Review · Reviewer_BduG · 2023-06-28

**Soundness:** 3 good
**Presentation:** 3 good
**Contribution:** 3 good
**Rating:** 6
**Confidence:** 3

**Summary:**

The authors propose a pre-trained deep learning wavefunction ansatz for variational Monte Carlo (VMC) solutions of the Schrödinger equation. The ansatz is trained on a chemically diverse set of (different conformations of) molecules. The zero-shot accuracy of the proposed method is a significant improvement over the previous state-of-the-art of pre-trained wavefunction ansätze, but additional fine-tuning steps are still necessary to obtain accurate relative energies. Due to the relatively simple model architecture, the achievable accuracy (in the limit of unlimited fine-tuning steps) is lower compared to more complex model architectures like PsiFormer, which however, need to be trained from scratch for every system.

**Strengths:**

The paper adresses the important problem of reducing the computational overhead of VMC methods using a deep learning wavefunction ansatz. The description of the proposed method is clear and well-written. Further, the authors include their code for reproducibility and the method is tested on a large number of systems.

**Weaknesses:**

While the paper compares results with previous methods, these comparisons consider only metrics like the number of optimization steps, but not the raw computational cost in terms of CPU/GPU time, which is arguably more important in practice. It is difficult to assess how "practical" the presented ansatz is compared to more traditional quantum chemistry methods.

**Questions:**

- Looking at the accuracy vs. number of heavy atoms curves shown in Figure 2, I wonder how "practical" the current method is compared to running coupled cluster calculations. For example, how expensive is the calculation of a system with six heavy atoms at the CC-4Z level compared to running 4k fine-tuning steps with the presented method? Are we at a stage already where VMC methods are comparable in cost, or even cheaper than conventional quantum chemistry methods?

- The authors show in Figure 3 and discuss in the text that the energies obtained with their method are not invariant w.r.t. global rotations. It is correct that the wavefunction is not invariant w.r.t. rotations, but it should be equivariant. The authors mention that they experimented with fully equivariant architectures, but observed worse results compared to their (non-equivariant) method. I assume this is because some ground state wavefunctions require "symmetry-breaking", e.g. the ground state of a carbon atom is not spherically symmetric. Still, once a "symmetry-broken" wavefunction is obtained, it should be equivariant under rotations. Have the authors already looked into the distinction between equivariance/symmetry-breaking? In my opinion, any insights regarding this problem are valuable and should probably be discussed in more detail (in the appendix/supplement).

- It is well known that the energy variance in VMC methods allows to assess how close the found solution is to the true ground state wavefunction, which would have a variance of 0. The authors show many plots with absolute/relative energies, but how do energy variances compare between different methods? Related to this: How large are energy variances when comparing the zero-shot to the few-shot setting?

- I believe there is a small mistake on line 122: Based on the description in the following lines ($l=0,\dots,L$ and $m=-l,\dots,l$), it should be  $x^{\mathrm{nuc}} \in \mathbb{R}^{N_\mathrm{nuc}\times(L+1)^2\times N_{\mathrm{channels}}}$ (instead of $x^{\mathrm{nuc}} \in \mathbb{R}^{N_\mathrm{nuc}\times(2L+1)^2\times N_{\mathrm{channels}}}$). Can you please check this?

**Limitations:**

The authors fairly discuss limitations of their work. While its potential for negative societal impact is not discussed, I think this is acceptable considering that this work is about the solution to a fundamental problem in quantum chemistry and a discussion of negative societal impacts would probably appear contrived.

---

> ### Author Rebuttal · Authors · 2023-08-09
>
> Thank you for your review! Regarding your specific comments:
>
> ### Weaknesses
>
> > While the paper compares results with previous methods, these comparisons consider only metrics like the number of optimization steps, but not the raw computational cost in terms of CPU/GPU time, which is arguably more important in practice. It is difficult to assess how "practical" the presented ansatz is compared to more traditional quantum chemistry methods.
>
> Like all other deep-learning-based VMC (DL-VMC) methods, our method is computationally costly. Computing the energy of a single molecule with 5 heavy atoms requires 10.5 node-hours with CCSD(T)-4Z (1 node = 256 CPU cores) and Psiformer requires 53h (for 16k epochs on 2xA100). Using our method requires 1h for zero-shot evaluation or 9.5h for 4k fine-tuning steps + evaluation (on 2xA100). This is to the best of our knowledge the first study that achieves similar accuracy at computational cost comparable to established quantum chemistry methods such as CCSD(T)-4Z. We will include these findings in the appendix of the final manuscript to clarify how the different methods compare with respect to computational cost.
>
> Furthermore a very recent preprint by Li. et al [3] has demonstrated that DL-VMC can be sped up by more than an order of magnitude, via a faster evaluation of the Laplacian operator in loss calculation. These per-step speed-ups can in principle be directly combined with our reduction in the required number of steps to yield a method that is competitive with established methods.
>
> Overall, our approach might not yet be practical compared to decades of quantum chemistry developments, but is already more accurate and efficient on several interesting systems.
>
> ### Questions
>
> > Looking at the accuracy vs. number of heavy atoms curves shown in Figure 2, I wonder how "practical" the current method is compared to running coupled cluster calculations. For example, how expensive is the calculation of a system with six heavy atoms at the CC-4Z level compared to running 4k fine-tuning steps with the presented method? Are we at a stage already where VMC methods are comparable in cost, or even cheaper than conventional quantum chemistry methods?
>
> In Table 1 of the rebuttal PDF, we have incorporated the timings for CCSD(T)-4Z, our zero-shot approach, and our approach with 4k fine-tuning steps (see also the response to weaknesses and the comment to all reviewers). In essence, when considering a molecule with 5 heavy atoms, the computational cost for our work, specifically with 4k fine-tuning, are on par with those of CCSDT(T), yielding equivalent levels of accuracy and runtime.
>
> > The authors show in Figure 3 and discuss in the text that the energies obtained with their method are not invariant w.r.t. global rotations. It is correct that the wavefunction is not invariant w.r.t. rotations, but it should be equivariant. The authors mention that they experimented with fully equivariant architectures, but observed worse results compared to their (non-equivariant) method. I assume this is because some ground state wavefunctions require "symmetry-breaking" [...]. Have the authors already looked into the distinction between equivariance/symmetry-breaking? In my opinion, any insights regarding this problem are valuable and should probably be discussed in more detail (in the appendix/supplement).
>
> We have added a comment on equivariance in the overall author rebuttal and will expand on this topic in the appendix of the revised manuscript.
> Beyond the challenges of equivariant wavefunctions described there, we would like to comment on the reviewer’s suggestion of obtaining a symmetry broken wavefunction that rotates with the molecule. This has been tried in the form of equivariant coordinate systems: Instead of using E(3)-equivarient networks, one canonicalizes the input to the network, by choosing a coordinate system which rotates with the molecule. This has been proposed by Gao et. al 2022 [1] (in the form of a global coordinate system, which rotates with the molecule) and Gerard et al. 2022 [2] (in the form of local atom centered coordinate systems, rotating with their local environments). However, despite ensuring equivariance, these methods have not been used in follow-up works by the respective groups. This is because these coordinate systems / symmetry-breakers change discontinuously as a function of the molecular geometry and can thereby introduce artificial discontinuities / kinks in the energy.
> We are unfortunately not aware of a method that can obtain symmetry broken, antisymmetric wavefunction that change smoothly as a function of geometry.
>
> > It is well known that the energy variance in VMC methods allows to assess how close the found solution is to the true ground state wavefunction, which would have a variance of 0. The authors show many plots with absolute/relative energies, but how do energy variances compare between different methods? Related to this: How large are energy variances when comparing the zero-shot to the few-shot setting?
>
> Fig. 1 of the rebuttal PDF, shows variances for all curves from Fig. 2 of the main text and we will include it in the final version.
> We observe a similar behavior as for the ground-state energy, wherein our approach exhibits very low variance even without fine-tuning. In contrast, other DL-VMC methods display a much higher initial variance and eventually converge to a similar point compared to our work after 32k optimization steps.
>
>
> > I believe there is a small mistake on line 122: Based on the description in the following lines (l=0, ..., L  and m=-l, ..., l) it should be x^{nuc} R^{N_nuc x (Lx1)^2 x N_channels} instead of x^{nuc} R^{N_nuc x 2(Lx1)^2 x N_channels}. Can you please check this?
>
> Thank you for pointing this out, we will fix it in the final manuscript.
>
> [1] Gao et al., arXiv:2110.05064, ICLR (2022)
>
> [2] Gerard at al., arXiv:2205.09438, NeurIPS (2022).
>
> [3] Li et al., arXiv:2307.08214, (2023).

---

> > ### Comment · Reviewer_BduG · 2023-08-13
> >
> > I thank the authors for their detailed reply. Further, I want to clarify that my comment regarding "practicality" was not meant negatively. I am aware that neural VMC methods are only an emerging field and have not received decades of optimisations like more well-established quantum chemistry methods. In this light, I find it quite impressive that the presented method has a relatively low computational cost and I encourage the authors to discuss computational cost more prominently in the final version of their manuscript. It is simply an important practical consideration with regards to wide-range adaption of these new methods in the quantum chemistry community.

---

> > > ### Author Response · Authors · 2023-08-14
> > >
> > > Thank you once again for your feedback and your help to improve the manuscript. We agree with the stated comment of the reviewer and will discuss more prominently the results regarding the computational cost in the final manuscript.
> > >
> > > Given the appreciation of our work, as well as the fact that we addressed all review comments in detail (clarified practicality, added computational cost, added experiments regarding energy variance, added discussion of E(3)-equivariance, fixed notation), we would like to encourage the reviewer to consider raising the final score to reflect these improvements.

---

### Official Review · Reviewer_R7Vm · 2023-07-06

**Soundness:** 2 fair
**Presentation:** 3 good
**Contribution:** 3 good
**Rating:** 5
**Confidence:** 4

**Summary:**

This paper concentrates on applying deep learning techniques to electronic structure computations. Unlike the strategy adopted in [1], the authors introduce a pretraining process to enhance the few-shot or zero-shot performance of Deep Learning Variational Monte Carlo (DL-VMC) methods. As their initial step, they design a novel architecture for the conventional model. Drawing inspiration from [2], the paper proposes a learnable orbital model for adjusting to molecular structures with a disparate number of electrons. The orbital model employs a neural network to predict the Fock and overlap matrices in the initial stages, and subsequently, it resolves a generalized eigenvalue problem to derive the orbital coefficients. Additionally, the network within the orbital model predicts the energy and selects new configurations for training. With the improved architecture, the authors pretrain the entire model using a dataset containing 98 distinct molecules. The pretrained model is benchmarked under various metrics, including absolute energy and relative energy. Furthermore, the authors conduct extensive experiments, comparing their zero-shot performance with Density Functional Theory (DFT) methods on the QM7 dataset. The empirical results demonstrate the effectiveness of their methodologies.

[1] David Pfau et al., Ab Initio Solution of the Many-Electron Schrödinger Equation with Deep Neural Networks

[2] Nicholas Gao and Stephan Günnemann, Generalizing Neural Wave Functions


**Strengths:**

1. Improved architecture. Compared to the [1, the authors employ a refined architecture to extract embedding features from electrons and nuclei. Rather than using orbital coefficients derived from the Hartree-Fock method, they directly predict the Fock matrix and overlap matrix, subsequently solving the generalized eigenvalue problem. This strategy eliminates the reliance on the Hartree-Fock method, an inaccurate solution for electronic structure. Ablation studies reveal that this enhanced structure significantly improves absolute energy performance in both zero-shot and few-shot settings.
2. Enhanced Sampling Methods. Despite the insignificance of the MCMC burn-in step in the standard FermiNet-like training process, it can be time-consuming in few-shot/zero-shot settings. To counteract this issue, the authors refine the initialization of MCMC walkers, achieving equilibrium within a shorter MCMC burn-in stage. Beyond this initialization enhancement, they also identify and address the burn-in requirements during the pretraining stage by adjusting the movement of electrons relative to the shifts in nuclei position. Both these strategies improve the convergence of MCMC walkers, leading to superior training performance.
3. Larger Pretraining Dataset. In addition to the technical enhancements, the authors also construct a more substantial dataset for pretraining, a development that could prove beneficial for future research in this area.
4. Extensive Empirical Study. The authors execute extensive experiments to showcase the effectiveness of their methods, including comparisons with the Density Functional Theory (DFT) method on a large-scale dataset.

[1] Michael Scherbela, Leon Gerard, and Philipp Grohs, Towards a Foundation Model for Neural Network Wavefunctions.

**Weaknesses:**

1. The relative energy experiments may not be convincing. Unlike the absolute energy, if the relative energy is not converged, the discrepancy between CCSD(T) and the proposed method could magnify with additional finetuning steps. Although model with 4k finetuning steps achieves relative energy nearly identical result as CCSD(T),  the reviewer suggests the authors to supply the relative energy result of model with an increased number of finetuning steps (e.g., 8k and 16k) to demonstrate energy convergence.
2. The comparison with Density Functional Theory (DFT) methods should also encompass speed comparison. Typically, Deep Learning Variational Monte Carlo (DL-VMC) methods should benchmark their energy results against post-HF methods and Quantum Monte Carlo (QMC) results, given that the total computation cost of these methods is comparable. Despite this paper's focus on zero-shot performance, the computational cost of evaluation of DL-VMC remains high. Therefore, the reviewer suggests that the authors provide speed comparisons to elucidate the effectiveness of their method.

**Questions:**

Questions about experiments have been shown in the weakness.

Questions about methods:
1. Why do the authors use the Hessian matrix of predicted energy to guide the sampling of configuration?
2. How to train the network to predict $E_{phis}$?
3. Why did the authors choose to solve the generalized eigenvalue problem to produce orbital coefficients? How is the gradient computed through this block?

**Limitations:**

Compared to other ab-initio methods, the proposed method requires pretraining on a large molecule dataset. Therefore, its performance highly depends on the similarity between pretraining and evaluation datasets, which limits the application of the method.

---

> ### Author Rebuttal · Authors · 2023-08-09
>
> Thank you for your feedback!  Regarding your specific questions and potential weaknesses:
>
> ### Weaknesses
>
> **W1**
>
> We ran additional experiments, increasing the nr of fine-tuning steps up to 32k, and depict the result in Fig. 2 of the rebuttal.
> We find that with increasing nr of fine-tuning steps, our relative energies converge towards the high-accuracy reference calculations. Furthermore, we wish to emphasize that we attain comparable results to the reference methods, utilizing a significantly lower number of fine-tuning steps per geometry (100k vs 3k for N2 and 64k vs. 3k for propadiene).
>
> **W2**
>
> Unfortunately, the reference we used for the DFT energies does not include run-times for the DFT calculations. Generally, DFT, being a mean-field solution, is typically significantly faster than Quantum Monte Carlo (including deep-learning-based methods), but also often much less accurate.
> The essence of this experiment was to provide a glimpse into the potential of our proposed approach, enabling rapid and precise computation of extensive datasets. Additionally, inclusion of this experiment within the manuscript serves to underscore the remaining challenge of extrapolating to substantially larger molecules, i.e. going from 4 heavy atoms during pre-training to 7 heavy atoms in the experiment. It's noteworthy that in the case of 4 heavy atoms, our method outperforms DFT calculations in terms of accuracy (and even CCSD(T) as seen in Fig. 2). However, when going to larger system sizes the zero-shot accuracy of our model decreases, as is clearly acknowledged as a current limitation of this model.
> To give a better sense of computational cost, we present a comparison of computational costs against CCSD(T) (cf. Table 1 of rebuttal pdf) and achieve similar accuracy with a similar amount of computational resources.
>
> ### Questions
>
> **Q1**
>
> Different degrees of freedom (DoF) in a molecule can have a strongly varying impact on the total energy. Some DoF (such as bond lengths) are typically “stiff”, i.e. small changes lead to large changes in energies. Others (such as twisting a single bond) are typically “soft”, i.e. have little impact on the energy.
> Molecules will therefore typically see large changes along soft DoFs, and only small changes for stiff DoFs. Preconditioning the random perturbations with the inverse of the Hessian achieves exactly that.
> Intuitively, instead of making perturbations of a given magnitude in geometry, this methods leads to perturbations of a given energy scale.
> We will expand Sec. 2.2 of the revised manuscript accordingly.
>
> **Q2**
>
> We train PhisNet to optimize for multiple objectives, one objective is directly to minimize the energy of a reference method, which is in our case a Hartree-Fock calculation. As it is common in supervised ML potential we predict the energy per atom from the node features and use as an estimator for the energy the sum over all atom contributions.
> We will add a clarifying sentence in the main text under section 2.3. and have a more detailed description of the loss function in Appendix F.
>
> **Q3**
>
> This is an established “trick” used e.g. by architectures such as SchNOrb or PhisNet, to obtain a well defined learning problem.
> Mean field orbital coefficients arise as solutions to an eigenvalue problem and are thus not uniquely defined. Eigenvectors are only defined up to their sign and for degenerate eigenvalues only up to an arbitrary rotation in the degenerate subspace. Furthermore, when looking for the eigenvectors corresponding to the lowest n eigenvalues (e.g. corresponding to the occupied orbitals), infinitesimal changes in the geometry can lead to different eigenvalues being included. The set of n lowest eigenvalues thus changes discontinuously as a function of geometry.
> These problems make learning orbital coefficients directly an ill-posed problem. The matrices F and S on the other hand are well defined and continuous as a function of input geometry. All ambiguities described above only exist in the diagonalization procedure and thus do not influence the learning process of the network itself. A recent preprint comparing direct learning of orbitals vs. learning of the matrices F and S [6] finds the latter approach to be vastly superior.
> Differentiation through this linear algebra routine is well supported in deep learning frameworks such as JAX (similar to differentiation through the determinant). In our specific application, we do not differentiate through the solver, but keep the weights of the pre-trained PhisNet model fixed during wavefunction optimization to avoid complexity.
>
> ### Limitations
>
> Although, the mentioned claim about the similarity between pre-training and evaluation dataset holds for any machine-learning-based method, we want to highlight that the pre-training dataset already contains a diverse range of chemical structures, encompassing equilibrium geometries, various chemical conformers, broken bonds, and different types of bond orders. This aspect sets our approach apart from previous methods like PESNet [3, 4], Globe [2], DeepErwin [5], and Transferable Atomic Orbitals [1]. We have managed to enhance the accuracy of the first properly generalizing wavefunction, which had poor zero-shot accuracy in Scherbela et al.'s work [1], by two orders of magnitude.
> Furthermore, our research demonstrates that significant improvements can be achieved by increasing the size of the dataset. Based on this finding, we hypothesize that further enhancements in generalizability can be attained by considering an even larger number of molecules and employing a more advanced scheme for sampling torsional angles.
>
>
> [1] Scherbela et al., arXiv:2303.09949, (2023).
>
> [2] Gao et al., arXiv:2302.04168, ICML (2023).
>
> [3] Gao et al., arXiv:2110.05064, ICLR (2022).
>
> [4] Gao et al., arXiv:2205.14962, ICLR (2023).
>
> [5] Scherbela et al., doi.org/10.1038/s43588-022-00228-x, Nat. Comp. Sci. (2022).
>
> [6] Van Workum et al, 10.26434/chemrxiv-2023-lwj87, (2023).

---

> > ### Comment · Reviewer_R7Vm · 2023-08-14
> >
> > Thank you for your reply. Your explanation of the method is very clear. However, the Propadiene torsion results in the additional file is confusing. In the main text, the authors claim that the discrepancy between the 4k finetune step result and CCSD(T) is within 2mha. In the additional file, the 0.4k finetune step result is close to CCSD(T) but the more finetuning step (3.2k) leads to an agreement with Psiformer. Which one is the correct finetuning step? Also, it would be beneficial to discuss which relative energy result should be considered as the ground truth in this problem.

---

> > > ### Author Response · Authors · 2023-08-14
> > >
> > > Thank you for the feedback and taking the time to assess our rebuttal. We are pleased that we managed to further clarify our method.
> > >
> > > **Regarding the nr of fine-tuning steps:**
> > > Both fine-tuning numbers are correct, but while we referred to the total nr of fine-tuning steps in the manuscript (i.e. 4000 steps in total, distributed among the 10 distinct twist angles), for the plot in the rebuttal we referred to the fine-tuning steps per geometry (0.4k = 4000 steps divided by 10 different twist angles). The plots are labeled correctly, but we should have consistently referred to the total nr of steps. We apologize for the confusion and will use consistent metrics in the final version of the manuscript. However, we do want to stress that our method uses at least an order of magnitude fewer steps per geometry than Psiformer on this problem.
> > >
> > > **Regarding energy ground-truth:**
> > > Prediction of relative energies as well as their assessment are notoriously difficult. Throughout this work we primarily compare against CCSD(T), since it is a well established, high-accuracy method, often regarded as “gold-standard” and used widely throughout computational chemistry.
> > > However, on a few systems CCSD(T) is known to potentially overestimate energy barriers. We believe that twisted propadiene might be one of these systems, hence we use Psiformer as ground-truth in the rebuttal figure. All 3 methods (CCSD(T), Psiformer, ours) qualitatively agree on this system, yielding energy barriers of 95-110 mHa. We do believe that on this specific system the results obtained by Psiformer and our method are correct and that CCSD(T) overestimates the barrier by ~15 mHa. We want to stress that the Psiformer calculations used 64,000 steps per geometry with twice our batch size, while even our most accurate fine-tuning calculations only required 3,200 steps per geometry. For our final manuscript we will add the estimated Psiformer barrier energy to Fig 3d and appropriately discuss the discrepancy between deep-learning-based-VMC and CCSD(T).

---

> > > > ### Comment · Reviewer_R7Vm · 2023-08-15
> > > >
> > > > Thank you for your reply. However, the reviewer believes the additional results do not fully address the concerns in weakness part. The experiments show that the relative energy can have 10mHa difference with large finetune steps, so it's hard to claim that the relative energy result is converged for 0.4k finetune step. As a result, the reviewer will not change the score.

---

> > > > > ### Author Response · Authors · 2023-08-16
> > > > >
> > > > > Thank you very much for engaging in detail with our work!
> > > > > We would like to respectfully challenge your assessment of our work.
> > > > >
> > > > > First of all, this work demonstrates substantial improvements on many different aspects of transferable neural wavefunctions, such substantially improved absolute energies and the regime of zero-shot and few fine tuning steps.
> > > > >
> > > > > Relative energies – being the small difference between two very large absolute energies – are a notoriously hard problem, but even on this aspect our method yields substantial improvements over the state of the art.
> > > > > To demonstrate this, we just reran PsiFormer with the same number of fine-tuning steps per geometry as our method and compare results in the table below (even using twice the batch-size for PsiFormer compared to our method). Even on this challenging task we obtain substantial improvements, obtaining 3-5x lower energy errors than PsiFormer. We do not claim in our paper to have solved the problem of relative energies (and have deliberately picked challenging systems in Fig. 3 to show the limitations of our method), but we do believe that our method actually performs very well compared to existing deep-learning-based VMC approaches.
> > > > >
> > > > > Barrier energy of propadiene twist:
> > > > > | Steps per geom. | $\Delta E$ our work | $\Delta E$ PsiFormer| Improvement |
> > > > > | -------- | ------- | ------- | ------- |
> > > > > | 1600 | 3.7 mHa   | 19.4 mHa | 5.2x |
> > > > > | 3200 | 2.8 mHa   | 8.3 mHa |  3.0x|
> > > > >
> > > > > Combined with the many other improvements demonstrated in a plethora of experiments, we are confident that this work will be of keen interest to the community and thus deserves being published at NeurIPS.

---

### Official Review · Reviewer_37BZ · 2023-07-12

**Soundness:** 3 good
**Presentation:** 3 good
**Contribution:** 3 good
**Rating:** 6
**Confidence:** 3

**Summary:**

The paper proposes a meta-learning model for neural wavefunctions. The neural model is first pre-trained by solving the variational optimization problem on many molecules. To adapt the model for a new problem (such as a new molecule), relatively few fine-tuning steps are needed to arrive at an accurate solution. The method consists of an orbital model based on PhisNet (with SE-3 equivariance), an improved electron embedding using message passing on the nuclear embeddings from PhisNet , and a Slater determinant to ensure antisymmetric property of the wavefunciton. The model is trained on a dataset of 98 small molecules with up to 4 heavy atoms. The resulting "base model" shows stronger zero-shot transfer across different number of heavy atoms than recent meta-learning wavefunction models. With a small amount of fine-tuning (4k steps), the model achieves very accurate energies, outperforming CCSD(T)-4Z on molecules up to 5 heavy atoms. It also achieves excellent accuracy on challenging systems like conformer energies, dissociation curves, and relative energies of transition states.

**Strengths:**

- Use of the improved architecture embeddings based on PhisNet that preserves SE-3 equivariance.
- The sampling of initial electron positions to run MCMC for VMC are improved by using an exponential distribution from the nuclei.
- For sampling geometries of the molecules, the covariance of the Gaussian is rectified by inverse Hessian of the energy, yielding better geometries that make more sense physically.

**Weaknesses:**

- Relative energies, which depend on small energy differences, are not very accurate without fine-tuning on that specific system. The transferability is not as good for relative energies.
- The overall end-to-end training is built upon previous works.

**Questions:**

- During sampling of molecular geometries, are they non-physical geometries discarded?
- Are the electron embeddings SE-3 equivariant? (I assume the MPNN used does not preserve equivariance.)

**Limitations:**

N.A.

---

> ### Author Rebuttal · Authors · 2023-08-09
>
> We thank the reviewer for the review and feedback regarding our submission.
>
> ### Weaknesses
>
> > Relative energies, which depend on small energy differences, are not very accurate without fine-tuning on that specific system. The transferability is not as good for relative energies.
>
> Predicting accurate relative energies remains a challenging task for DL-VMC methods. Previous studies have highlighted a noticeable discrepancy between relative energies predicted by DL-VMC and conventional high-accuracy quantum chemistry approaches. For instance, in the case of the studied Nitrogen dimer, several high-accuracy DL-VMC methods predict an energy barrier for bond breaking that is several mHa higher than what is observed in experiments and conventional approaches [3]. Additionally, for the Benzene dimer, despite investing several thousands of GPU hours, DL-VMC fails to accurately predict the sign of the relative energies [2, 4].
> Another relevant work, which also optimizes a wavefunction across multiple molecules simultaneously, overestimated the energy barrier for Ethene by several 100mHa's [1, 5]. While DL-VMC generally achieves state-of-the-art accuracy in absolute energies, it falls short in providing accurate predictions for specific systems' relative energies. Nonetheless, we openly and transparently acknowledge this limitation in our limitations section and anticipate that future research will address and improve upon this aspect.
>
> > The overall end-to-end training is built upon previous works.
>
> First of all, we believe that building on good previous work is admirable rather than deplorable. However, we do add several novelties that yield our substantially improved results:
>
> - The original PhisNet was only trained on MD simulations of a single molecule, which is insufficient in our case. We therefore pre-train PhisNet across a variety of chemically diverse systems simultaneously. To enable this we made several simplifications to PhisNet to increase stability as listed in Appendix F.
>
> - We improved the architecture of the electron embedding, lowering energy errors by 50 %.
>
> - We added a sophisticated sampling strategy to obtain a diverse but physically plausible set of molecule geometries.
>
> - We successfully scale training to this more diverse pre-training dataset, allowing us to reach two orders of magnitude better zero-shot performance to previous work.
>
>
> ### Questions
>
> > During sampling of molecular geometries, are they non-physical geometries discarded?
>
> We avoid non-physical geometries by distorting the geometries with respect to the normal modes of the Hessian of the energy. Furthermore, we bias each distortion towards the original geometry and reset the distortion trajectory every 20 epochs to the origin. Apart from that we do not reject any geometries and don’t observe instabilities with respect to unphysical geometries.
>
> > Are the electron embeddings SE-3 equivariant? (I assume the MPNN used does not preserve equivariance.)
>
> The electron embeddings are indeed not SE(3)-equivariant. We deliberately break this symmetry by using cartesian coordinates as input features, similar to all other neural wavefunctions such as FermiNet, Psiformer, DeepErwin or PESNet.
> This symmetry breaking is required, because the orbitals used to build the wavefunction must be symmetry-broken in order to be sufficiently expressive. If the orbitals are built from SE(3)-equivariant features without symmetry breaking, the resulting wavefunction is generally invariant to rotations [6, appendix B] and thus insufficiently expressive.
> To obtain approximately rotation invariant energies from this symmetry-broken wavefunction, we use data augmentation during pre-training of our wavefunction, by randomly rotating geometries every 20 steps as described in Sec 2.2 (l. 174-175).
> To obtain truly rotation invariant energies from this approximately invariant model, we can also apply random rotations during inference. Because the evaluation energy is a Monte Carlo average over many different electron samples, we can choose different global rotations for these samples at no additional computational cost, to obtain rotationally averaged (and thus invariant) energies.
> We demonstrate this technique on the example of a globally rotated propadiene molecule in the rebuttal document, Fig. 3.
>
> [1] Scherbela et al., arXiv:2303.09949, 2023.
>
> [2] Von Glehn et al., arXiv:2211.13672, ICLR (2023)
>
> [3] Gerard et al., arXiv:2205.09438, NeurIPS (2022).
>
> [4] Ren et al., doi.org/10.1038/s41467-023-37609-3, Nature Communications (2023).
>
> [5] Gao et al., arXiv:2302.04168, ICML (2023).
>
> [6] Gao et al., arXiv:2110.05064, ICLR (2022)

---

> ### Author Response · Authors · 2023-08-17
> **Discussion ends soon**
>
> Thank you once again for your feedback and the help to improve our work.
>
> Since the discussion ends soon, we would highly appreciate your feedback on our rebuttal.
> You might be particularly interested in:
> - An improved method to obtain rotation invariant zero-shot energies at inference
> - Additional experiments on convergence of relative energies
> - Our favorable comparison of computational cost vs. CCSD(T)
> - Analysis of the convergence of energy variance, indicating high wavefunction quality
>
> In case we addressed your concerns, we would appreciate it if you considered reflecting these improvements in your score.

---

> > ### Comment · Reviewer_37BZ · 2023-08-17
> >
> > Thanks for the response and clarifications. I don't have any concerns at the moment and will keep my original rating given the  strengths and weaknesses of the paper.

---

### Official Review · Reviewer_NURi · 2023-07-16

**Soundness:** 3 good
**Presentation:** 2 fair
**Contribution:** 2 fair
**Rating:** 5
**Confidence:** 3

**Summary:**

This paper tackles the problem of solving quantum Monte Carlo optimization on multiple molecules. A PhisNet model is used to encode geometric information from nuclei and other GNN models are then used to embed electrons. Additionally, the authors propose a method to generate more diverse nuclear geometries by a specially designed distortion method. Experiments show that the method can achieve good accuracy when trained smaller molecules and test on larger molecules. After some fine-tuning, the method can achieve improved results.


**Strengths:**

- The authors provide detailed description of the architecture.
-  The paper uses diverse datasets to evaluate their method.

**Weaknesses:**

- The improvements of the proposed method mainly comes from an improved architecture. To my understanding, the novelty seems to be limited.
- The proposed method does not yield improved results when more optimization steps are used.
- The presentation of the training and evaluation steps is not clear. Specifically, how is pre-training and fine-tuning done.

**Questions:**

- Among the many experimented datasets, which one(s) most decisively show the benefit of the proposed method?

---

> ### Author Rebuttal · Authors · 2023-08-09
>
> We thank the reviewer for the feedback and the effort to improve our work. We hope we can address in the following rebuttal all raised questions and comments.
>
> ### Weaknesses
>
> > The improvements of the proposed method mainly comes from an improved architecture. To my understanding, the novelty seems to be limited.
>
> The architecture yields roughly 50% of the improvements, which we explicitly demonstrate in the ablations (Fig. 4). Other important contributions to the performance are for example the initialization of Monte Carlo walker, significantly reducing the number of necessary burn-in steps as highlighted in Appendix B (Fig. 6), which is a large part of the computational cost when performing zero-shot evaluations or the continuous sampling of geometries with normal mode distortion during pre-training of a base model.
>
> > The proposed method does not yield improved results when more optimization steps are used.
>
> Our method does yield improved results with more optimization steps, both during pre-training and during fine-tuning: For pre-training we show in Fig. 4 that increasing the number of pre-training steps from 128k to 256k yields large gains in accuracy of 37% for zero-shot and 21% for fine-tuning respectively. For fine-tuning Fig. 2c shows continuing improvements in energy, and our rebuttal Fig. 1 shows corresponding improvements in energy variance.
> It is true that during fine-tuning we see diminishing returns with more fine-tuning, but this can be primarily ascribed to already starting from a very good wavefunction right after pre-training.
> Furthermore, as shown in Fig. 2, we outperform other generalized wavefunctions by two orders of magnitude in the zero-shot regime. We are able to reach the same performance of other generalized wavefunctions with 8x less optimization steps and surpass this accuracy with additional steps (cf. Fig 2c blue vs. orange). Additionally, we also outperform well established high-accuracy methods such as CCSD(T) with very few fine-tuning steps, again demonstrating a large improvement over the previous SOTA.
>
> > The presentation of the training and evaluation steps is not clear. Specifically, how is pre-training and fine-tuning done.
>
> Thank you for the feedback. We will add an additional clarifying paragraph to the main text in section 2.3 and highlight the specific difference to pre-training and fine-tuning a neural wavefunction for the final version of the manuscript.
> To summarize in the present manuscript we refer to pre-training as the stage, where a neural wavefunction is optimized extensively on a number of molecules (in our case up to 98 molecules) with varying number of geometries. This pre-trained wavefunction is used to initialize a wavefunction for a potentially completely new molecule for further fine-tuning steps. Normally, the fine-tuning stage is rather short with a couple of hundred optimization steps (vs. thousands of steps in a standard calculation). When no additional optimization steps are performed and the wavefunction is directly evaluated we speak of zero-shot predictions.
>
>
> ### Questions
>
> > Among the many experimented datasets, which one(s) most decisively show the benefit of the proposed method?
>
> The most direct comparison is against the work by Scherbela et al.. We are able to significantly improve upon their zero-shot results by two orders of magnitude (Fig. 2a). We also outperform CCSD(T) calculations for systems significantly larger than the ones present in our training set. The fine-tuning accuracy is improved by a factor of 5x in the extrapolation regime of 5 heavy atoms (Fig 2b). Additionally, we reach the same accuracy as Scherbela et al. for 3 heavy atoms 8x faster with a significantly lower final ground-state energy prediction (cf. Fig 2c).
> Moreover, our approach demonstrates competitive performance even against DL-VMC methods tailored for specific systems, requiring an order of magnitude fewer optimization steps, as evidenced in the bicyclobutane experiment (cf. Fig 3a).

---

> ### Author Response · Authors · 2023-08-17
> **Discussion ends soon**
>
> Thank you once again for your feedback and the help to improve our work.
>
>
> Since the discussion ends soon, we would highly appreciate your feedback on our rebuttal.
> We believe to have extensively addressed your concerns regarding novelty, convergence with more optimization steps, and clarity of method description.

---

> > ### Comment · Reviewer_NURi · 2023-08-17
> >
> > Thanks for the rebuttal. Although I do not observe a significant leap in the methodology part, contributions such as improved architecture, larger dataset and improved MCMC burn in, are of value. I appreciate the detailed empirical study and will increase my score. Another comment for the presentation is that the authors use multipliers like 300x or 5x extensively in expressing the improvements in performance. However, such multipliers could be difficult to parse or show the meaningfulness of the results partly due to the lack of standardized comparison target.

---

### Official Review · Reviewer_m8hG · 2023-07-24

**Soundness:** 4 excellent
**Presentation:** 3 good
**Contribution:** 4 excellent
**Rating:** 7
**Confidence:** 4

**Summary:**

The paper proposes a neural network wave function model which generalizes to different molecules and geometries. The neural network includes electron model, orbital model, both with message passing, Slater determinants, and an energy estimate for geometry sampling. The network is pretrained on 98 molecules and results show lower energies with one-shot and fine-tuning, on new molecules, compared to other generalized wavefunctions and some computational chemistry approaches. The paper also includes an ablation study and prediction of absolute energies for 250 molecules.

**Strengths:**

The idea is original since there are very few methods of generalizable neural networks for wavefunctions.
The normal mode of sampling geometries from an inverse hessian of energy seems to be original.
The experiments and figures are high quality.
The results are very good, it seems to be the first generalizable, ab initio quality wavefunction, that can be trained with quantum Monte Carlo. Generalizability can be computationally efficient if not every optimization needs to be started from scratch.
The problem of solving Schrodinger equation is an important and significant one.

**Weaknesses:**

I'm not so familiar with CC-2Z, -3Z etc methods the paper compares against. Not sure how the proposed method compares with something like Slater-Jastrow variational Monte Carlo (which is quite fast and generalizable for an abinitio method).

**Questions:**

From Fig. 3c, it seems that the errors are somewhat dependent on the rotation angle. Is that actually happening?

How much of the improvement attributed to "improved architecture" may be due to Phisnet?

**Limitations:**

The authors are upfront about the limitations: potential limited expressiveness with more optimization iterations, prediction is worse for increasing extrapolation and out-of-distribution predictions.

---

> ### Author Rebuttal · Authors · 2023-08-09
>
> Thank you very much for your review! Regarding your specific questions and potential weaknesses:
>
> ### Weaknesses
>
> >I'm not so familiar with CC-2Z, -3Z etc methods the paper compares against. Not sure how the
> proposed method compares with something like Slater-Jastrow variational Monte Carlo (which is quite fast and generalizable for an abinitio method).
>
> When we refer to CC-2Z, etc. we refer to CCSD(T), Coupled Clousters Single Doubles (Perturbative Triples), which is the workhorse of high-accuracy calculations in computational chemistry. A key choice in these calculations is the basis size, for which typically correlation consistent basis sets are being used. Their size is typically denoted by double-zeta, triple-zeta, etc. and thus often abbreviated as 2Z, 3Z, etc. Thus when we refer to CC-4Z, we refer to CCSD(T) with the basis set ppCV4Z. We will explain this in more detail in the updated manuscript.
> When comparing CCSD(T) against Slater-Jastrow Monte Carlo, CCSD(T) is typically vastly superior in energies, but also computationally much more expensive. A systematic comparison of SJ-VMC and SJ-DMC vs. neural wavefunctions can for example be found in Fig. 7 of [1], showing that SJ-VMC is typically ~100 mHa higher in energy and SJ-DMC ~10 mHa higher in energy.
> As simple test, we took a random molecule (H2CO) from a SJ-DMC benchmark [2] and compared energies obtained from SJ-DMC, CCSD(T) and our 0-shot wavefunction. We also find CCSD(T) to be superior to SJ-DMC, only outperformed by neural wavefunctions:
> | Method                | Energy / mHa |
> |-----------------------|--------------|
> | SJ-DMC                | -114.480     |
> | CCSD(T)-4Z            | -114.485     |
> | Our 0-shot evaluation | **-114.488**     |
>
> ### Questions
>
> >From Fig. 3c, it seems that the errors are somewhat dependent on the rotation angle. Is that actually happening?
>
> Yes, because our wavefunction is not fully SE(3)-equivariant, the obtained energies are not rotation invariant, and thus depend on the chosen global rotation angle. Currently no architecture is known that allows SE(3)-equivariance of the wavefunction, efficient antisymmetrization via the determinant and smooth dependency on the geometry at the same time. We will add further details on this point in the appendix of the revised manuscript.
> Having said that, our energies are approximately rotation invariant, because we randomly perturb and rotate geometries during variational pre-training, leading to similar energies for any global rotation angle.
> From such an approximately rotation-invariant model we can even obtain fully rotational invariant energies, as depicted in the rebuttal Fig. 3. By also applying random rotations during evaluation (which requires no substantial extra cost), we can obtain fully rotational invariant energies (up to Monte Carlo noise).
>
> > How much of the improvement attributed to "improved architecture" may be due to Phisnet?
>
> PhisNets orbital descriptors do not contribute to the accuracy directly. PhisNet does have an implicit effect on the accuracy, by being able to efficiently predict the Hessian matrix and therefore being able to enhance sampling of the geometries it contributes to improved accuracy.

---

> > ### Comment · Reviewer_m8hG · 2023-08-16
> >
> > I thank the authors for their reply. The points on SJ-DMC and rotation angle addressed my questions. Perhaps I'm misunderstanding why the PhisNets orbital descriptors do not contribute to accuracy directly as they seem to be inputs to the orbital embedding net and nuclear features from Fig. 1 of manuscript?

---

> > > ### Author Response · Authors · 2023-08-16
> > >
> > > The PhisNet model serves three distinct purposes in our method:
> > >
> > > 1. To generate orbital coefficients, which are used as input to the orbital embedding net
> > > 2. To generate nuclear features, which in turn affect the electron embeddings
> > > 3. To generate an energy estimates for the normal mode sampling
> > >
> > > Purpose 1 is essential, and the method would not work at all, without obtaining n_electron distinct orbital features, because these are required to build the n_electron x n_electron Slater determinant. In prior work on transferable wavefunctions this has been achieved using heuristic orbital locations (Gao et al.) or using an iterative mean-field method such as Hartree-Fock (Scherbela et al.). By introducing PhisNet, we can replace the heuristic or iterative Hartree-Fock method by a trainable neural network, which outputs the Hartree-Fock coefficients. In that sense PhisNet does not improve to accuracy gains beyond Scherbela et al., because it yields essentially the same outputs as Hartree-Fock, although substantially faster. This is the point we meant with “PhisNets orbital descriptors do not contribute to the accuracy directly.”.
> > >
> > > Purpose 2 is in principle optional, but since the nuclear embeddings are available for free as a by-product from PhisNet, we decided to use them. We did not quantify this contribution to accuracy separately.
> > > Intuitively we would expect these nuclear embeddings – which contain information about the chemical environment of each nucleus – to improve beyond a pure one-hot encoding of the nuclear charge.
> > >
> > > Purpose 3 has an effect via better sampling, improving fine-tuning accuracy by ~12%.

---

> > > > ### Comment · Reviewer_m8hG · 2023-08-21
> > > >
> > > > I see, Thanks for your reply!

---

### Author Rebuttal · Authors · 2023-08-09

We thank all reviewers for their constructive feedback and appreciate the overall positive reception of our work (“excellent accuracy on challenging systems”, “extensive Empirical Study”), as well as its presentation (“experiments and figures are high quality”, “clear and well-written”).

Several reviewers have asked for additional experiments, which can be found in the attached rebuttal pdf. We ran the following additional experiments, which we will include in the appendix of the revised manuscript:

- Fig 1: Analyzing the energy variance of our wavefunction instead of the mean energy (reviewer BduG) shows the same behavior: Our zero-shot evaluations obtain substantially lower variance than the previous zero-shot-SOTA.
- Fig 2: Fine-tuning of our model for more epochs (reviewer R7Vm), shows that our relative energies do indeed improve further with additional fine-tuning and converge to accurate values.
- Fig 3: Our model yields rotation invariant energies through averaging energies of randomly rotated geometries during inference without additional computational cost
- Tab 1: Run-time comparison. We find that our method is on par with CCSD(T) in terms of accuracy and speed, and can be ~5x faster than Psiformer.

We have answered each reviewer individually, and additionally comment on three recurring themes, which we will also highlight better in the final manuscript.

### Novelty

We believe our contributions to be substantial and novel in multiple regards: We obtain for the first time deep-learning-based VMC-energies for small molecules that are competitive with CCSD(T) in both accuracy as well as run-time. We improve energy errors for zero-shot evaluations by 300x (Fig. 2a) and improve energy errors for fine-tuned wavefunctions by 5x (Fig. 2b) compared to the previous SOTA. These large improvements are obtained through a plethora of contributions:
- Improved electron embedding architecture, reducing energy error by 50% vs Scherbela et al.
- Hessian-based sampling strategy, generating a chemically diverse but physically plausible pre-training dataset
- Iterative Hartree-Fock replaced by adapted version of PhisNet. While the original PhisNet was only trained on geometries of a single molecule, we manage to train on a diverse dataset through various stability improvements, detailed in Appendix F.
- Improved electron position initialization, to reduce burn-in time and computational cost
- Thorough analysis of strengths and limitations of the resulting model, establishing a strong benchmark for further research

### Computational cost

Like all other deep-learning-based VMC (DL-VMC) methods, our method is computationally costly. However, to our knowledge this is the first study that achieves high accuracy at computational cost comparable to established quantum chemistry methods such as CCSD(T). Computing the energy of a single molecule with 5 heavy atoms requires 10.5 node-hours with CCSD(T)-4Z and Psiformer requires 53h (for 16k epochs). Using our method requires 1h for zero-shot evaluation or 9.5h for 4k fine-tuning steps + evaluation.
We thus reach the same accuracy as CCSD(T)-4Z in the same amount of time, and are 6x faster than Psiformer.
Furthermore a very recent preprint by Li. et al [1] has demonstrated that DL-VMC can be sped up by more than an order of magnitude, via a faster evaluation of the Laplacian operator in loss calculation. These per-step speed-ups could in principle be directly combined with our reduction in the required number of steps, yielding even higher speed-ups.

### E(3)-equivariance

Like all other neural wavefunction architectures known to us (incl. Psiformer, DeepErwin, GLOBE), also our architecture is not equivariant.
While E(3)-equivariant networks have been successful in neural network potentials (e.g. Nequip, MACE), they appear inapplicable to neural wavefunctions. This is because the orbitals used to construct the Slater determinant can have lower symmetry than the Hamiltonian, i.e. be symmetry-broken. By design, E(3)-equivariant networks output functions which are symmetry compatible with the input and are thus not suited to model symmetry broken orbitals. Constructing the wavefunction from E(3)-equivariant orbitals, leads to an invariant wavefunction as shown in Appendix B of Gao et al 2022 [2].

Furthermore, Pozdnyakov and Ceriotti demonstrate in a concurrent pre-print [3] that highly accurate neural network potentials, which outperform the E(3)-equivariant SOTA, can be obtained without explicit E(3)-equivariance. They achieve this with a non-equivariant architecture and data augmentation during training. While E(3)-equivariance is a very elegant concept, it might ultimately not even be required to obtain accurate wavefunctions.

Our approach is thus as follows: All parts of the network, which purely depend on nuclear coordinates (ie. PhisNet) are equivariant. All parts of the network which depend on electron coordinates (requiring symm. breaking) are not equivariant. We obtain approximately rotation invariant energies through data augmentation, by randomly rotating geometries during variational pre-training. While this procedure does not yield fully rotationally invariant energies, our experiment on rotated propadiene (Fig. 3c) demonstrates that the energies are almost invariant (max. zero-shot deviation 5mHa) and improve via fine-tuning.

We can fully enforce energy invariance, by also using data augmentation during evaluation (rebuttal Fig 3). Instead of evaluating the energy for a fixed geometry, we can sample energies for randomly rotated molecules during inference. Since we require Monte Carlo sampling of many different electron configurations anyway, this comes at no additional computational cost and yields rotation invariant energies up to Monte Carlo noise.

[1] Li et al., arXiv:2307.08214, (2023).

[2] Gao et al., arXiv:2110.05064, ICLR (2022)

[3] Pozdnyakov et al., arXiv:2305.19302, (2023).

---

### Decision · Program_Chairs · 2023-09-21

**Decision:**

Accept (poster)

**Comment:**

This paper considers an approach to deal with the large computational cost of deep learning-base variational Monte Carlo (DL-VMC) methods. To do so, the paper proposes a DL-VMC model that is pre-trained on a large set of molecules and show that a small number of fine-tuning steps can lead to accurate energies in a number of test systems. While the reviewers have pointed out a number of limitations in this work, overall reviewers appreciated the empirical results of the paper and came to a consensus to accept the paper. For the final version, we strongly encourage the authors to revise the manuscript according to the reviewers' feedback.